# Characterization of fresh and aged organic aerosol emissions from meat charbroiling

Christos Kaltsonoudis[1,2], Evangelia Kostenidou[1], Evangelos Louvaris[1,2], Magda Psichoudaki[1,2],

Epameinondas Tsiligiannis[1,2], Kalliopi Florou[1,2], Aikaterini Liangou[1,2], and Spyros N. Pandis[1,2,3]

[1]Institute of Chemical Engineering Sciences, ICE-HT, Patras, Greece

[2]Department of Chemical Engineering, University of Patras, Patras, Greece

[3]Department of Chemical Engineering, Carnegie Mellon University, Pittsburgh, USA

*Correspondence to*: Spyros N. Pandis (spyros@andrew.cmu.edu)

**Abstract**. Cooking emissions can be a significant source of fine particulate matter in urban areas. In this study the aerosol and gas phase emissions from meat charbroiling were characterized. Greek souvlakia with pork meat were cooked using a commercial charbroiler and a fraction of the emissions were introduced into a smog chamber where after a characterization phase they were exposed to UV illumination and oxidants. The particulate and gas phases were characterized by a High-Resolution Time-of-Flight Aerosol Mass Spectrometer (HR-ToF-AMS) and a Proton-Transfer-Reaction Mass Spectrometer (PTR-MS) correspondingly. More than 99% of the aerosol emitted was composed of organic compounds, while black carbon (BC) contributed 0.3% and the inorganic species less than 0.5% of the total aerosol mass. The initial O:C ratio was approximately 0.09 and increased up to 0.30 after a few hours of chemical aging (exposures of $10^{10}$ molecules $cm^{-3}$ s for OH and 100 ppb hr for ozone). The initial and aged AMS spectra differed considerably ($\theta=27^0$). Ambient measurements were also conducted during Fat Thursday in Patras, Greece when traditionally meat is charbroiled everywhere in the city. Positive Matrix Factorization (PMF) revealed that COA reached up to 85 % of the total OA from 10:00 to 12:00 LST that day. The ambient COA factor in two major Greek cities had a mass spectrum during spring and summer similar to the aged meat charbroiling emissions. On the other hand, the ambient COA factor during winter resembled strongly the fresh laboratory meat charbroiling emissions.

## 1 Introduction

Organic aerosol (OA) is one of the main components of atmospheric particulate matter (PM) (Kanakidou et al., 2005; Zhang et al., 2007). Identification of the sources of OA has proven to be a difficult task due to their diversity and the continuous chemical evolution of the corresponding organic compounds. The Aerosol Mass Spectrometer (AMS, Aerodyne Research) provides continuous information (OA mass spectra) that allows the identification of some OA sources. Additionally, the OA elemental ratios (O:C, H:C, N:C) can be calculated providing useful information about the average chemical state of the OA (Aiken et al., 2008). Positive matrix factorization (PMF) (Paatero and Tapper,

1994; Lanz et al., 2007) is often used to deconvolute the AMS data into a linear combination of factors. The resulting OA factors are associated to primary OA (POA), such as the hydrocarbon-like organic aerosol (HOA) or oxidized OA (OOA) which in many cases has been related to secondary OA (SOA) (Zhang et al., 2007). Factors linked to biomass burning emissions (BBOA), cooking emissions (COA) and marine emissions (MOA) have also been identified. The OOA has been further separated into factors based on their degree of oxidation and volatility (Zhang et al., 2007; Kostenidou et al., 2009; 2015; Sun et al., 2011; Ge et al., 2012; Mohr et al., 2012; Crippa et al., 2013).

Cooking organic aerosol (COA) has been found to represent from 10 to 35% of the total OA measured in urban locations (Allan et al., 2010; Sun et al., 2011; 2012; Ge et al., 2012; Mohr et al., 2012; Crippa et al., 2013; Lee et al., 2015). In Greece the COA levels have been estimated in two major cities: Athens and Patras. During the summer the COA related source (named HOA-2) was 17% and 14% of the total OA in Athens and Patras respectively (Kostenidou et al., 2015). For the winter the corresponding contributions were 16% for Athens and 12% for Patras (Florou et al., 2016).

Emissions from meat cooking may produce large amounts of aerosol up to 40 g per kg (Hildemann et al., 1991). The types of meat cooked (chicken, beef etc.), other food ingredients or the cooking method affect both the aerosol emission rate and the composition of the corresponding particles (Rogge et al., 1991; Mohr et al., 2009; He et al., 2010). For example, Allan et al. (2010) suggested that the oil used during meat frying may contribute more to the emitted PM than the meat itself in urban areas in the United Kingdom.

Meat cooking particles contain palmitic acid, stearic acid, oleic acid, nonanal, 2-octadecanal, 2-octadecanol, and cholesterol (Rogge et al., 1991). Schauer et al. (2002) measured the emissions from cooking with seed oils, showing that this process is a source of n-alkanoic and n-alkenoic acids. Allan et al. (2010) reported AMS spectra for several oils used for cooking, showing similar spectra (with enhanced fractions of signal at m/z 41 and 55) with some COA factors reported in the literature.

Most of the AMS spectra from ambient measurements related to COA are characterized by peaks at m/z values 41, 43, 55, 57, 69, etc., and have an O:C ratio ranging from 0.08 to 0.21 (Mohr et al., 2009; 2012; Allan et al., 2010; He et al., 2010; Sun et al., 2011; 2012; Ge et al., 2012; Crippa et al., 2013; Hayes et al., 2013). He at al. (2010) reported coefficients of determination ($R^2$) of 0.95 – 0.98 among the spectra of OA emissions from different types of Chinese cooking, despite the differences in ingredients and cooking methods. Mohr et al. (2009) compared the spectra of OA produced by grilling of hamburgers and chicken without skin. $R^2$ values greater than 0.9 were found between these AMS spectra. Mobile aerosol measurements indicate that commercial and residential cooking contribute to to enhanced OA concentrations (Elser et al., 2016).

Despite the previous efforts, there are a number of remaining questions regarding the characterization of the emissions related to cooking practices. Separation of COA from the HOA and other primary components is still a challenge for the PMF analysis (Mohr et al., 2009; Kostenidou et al.,

2015). Ots et al. (2016) attempted to constrain COA emissions in the UK using the AMS-PMF results. Furthermore, the fate of these primary emissions in the atmosphere is still unknown. The reactions with ozone ($O_3$) and OH radicals may significantly alter these aerosols. Hearn et al. (2005) studied the reaction of oleic acid particles with ozone and concluded that relatively fast heterogeneous reactions occur at the surface of the particles. Dall'Osto et al. (2015) reported different COA factors for a rural site in the Po Valley, Italy with one being associated partially with primary organic aerosol components such as HOA and partially with secondary components. On the contrary, the second COA factor did not correlate with primary tracers. Kostenidou et al. (2015) reported an HOA-2 factor for the summer measurements in Athens and Patras, Greece that appeared to be associated with cooking but was quite different from the COA factor identified in winter in the same areas by Florou et al. (2016). The reasons for the differences of the COA factor spectra even if the cooking practices are the same in the two seasons were not clear. Due to the mild climate in Greece, there is no significant change in what is cooked during the different seasons as opposed for example to cities in much colder climates.

The aim of this work is to characterize the particulate emissions of pork meat charbroiling, an activity that is thought to produce large amounts of OA. Smog chamber experiments were conducted in order to characterize the fresh and aged meat charbroiling emissions. The resulting spectra were compared to COA factors derived from ambient measurements in Greece during different periods of the year in an effort to explain the apparent differences in COA spectra derived from the PMF analysis.

## 2 Experimental procedures

### 2.1 Chamber experiments

A set of five smog chamber experiments were conducted in the ICE-HT environmental chamber facility. This facility is composed of a temperature-controlled smog chamber room (3 m W x 4.5 m L x 2.5 m H) incorporating over 300 UV light lamps (Osram, L36W/73) capable of producing a $J_{NO2}$ of 0.6 min$^{-1}$ (when all lights are turned on). The reactor had a volume of 10 m$^3$. A commercial charbroiler was used for the meat cooking. Natural wood coal was purchased from local distributors. A butane burner was used for the ignition of the coal. The charbroiler was placed outside the laboratory and adequate time was allowed for the coal ignition. Pork meat was purchased from the local market. The meat was cut into 2x2x1 cm pieces which were placed on wood sticks (length: 20 cm). Approximately 100 g of meat were used for each souvlaki. This type of cooking is widely used in Greece, both at restaurants and homes. 10 – 15 souvlakia were cooked for approximately 20 minutes. Using a metal bellows pump (Senior Aerospace, model MB 602) a fraction of the emissions was introduced into the 10 m$^3$ Teflon (PTFE) chamber that had been pre-filled with clean air. Insulated 3/8 in copper tubing (less than 2.5 m in length) was used to transfer the cooking emissions into the chamber. The copper tubing used for the sampling was insulated and was therefore heated by the exhaust vapors. We have confirmed that the metal bellows pump, as expected based on its design, does not generate particles or VOCs. The PM$_1$

losses in this pump have been characterized previously (Kostenidou et al., 2013) using 2 SMPS systems for both ammonium sulfate and ambient particles. The losses were less than 10% for particles larger than 150 nm, increasing to 30% for 100 nm particles. The flow rate for the transfer line was approximately 170 L min$^{-1}$. The temperature and relative humidity were in the range of 20-25$^{o}$C and 15-35% respectively. Air was sampled 1 m above the charbroiler for approximately 10 min in order to achieve a concentration inside the chamber of the order of 100 - 500 μg m$^{-3}$. The conditions of each experiment are shown in Table 1. The NO$_x$ concentrations were in the range from 1.5 to 8 ppb. The NO$_2$ to NO ratio ranged from 2 to above 10. The NO$_x$ analyzer used is sensitive to other NO$_y$ compounds and thus its measurements represent an upper limit of NO$_2$.

A HR-ToF-AMS (Aerodyne Research Inc.) measured the non-refractory PM$_1$ aerosol. The vaporizer temperature was set at 600$^{o}$C and the voltage difference between the filament and the ion chamber was 70 V. The V mode of the AMS was used in these experiments. A Scanning Mobility Particle Sizer (SMPS, classifier model 3080, DMA model 3081, CPC model 3787, TSI) measured the particulate number size distribution. The sheath flow rate was 5 L min$^{-1}$ and the sample flow rate was 1 L min$^{-1}$. The size range of the SMPS under this configuration is from 10 to 500 nm. The 10:1 ratio provides more accurate size distribution measurements as the instrument has a sharper transfer function but it reduces the measurement range to 10-300 nm. Given the modest size accuracy requirements in this study (a few percent), we selected to cover a larger size range instead. A Multiple-Angle Absorption Photometer (MAAP, Thermo Scientific Inc.) was used for the measurement of the PM$_1$ particulate black carbon.

Quartz filters, placed after a PM$_{2.5}$ cyclone, were used to collect samples of the emitted COA from directly above the charbroiler and from inside the chamber at the end of selected experiments. These samples were used for the measurement of the organic (OC) and elemental carbon (EC) by thermal – optical analysis (Sunset Laboratory Inc., EUSAAR 2 protocol) and for the analysis of the water soluble organic carbon (WSOC). For the WSOC extraction a *P* parameter value equal to 0.1 cm$^3$ m$^{-3}$ was used according to Pscichoudaki and Pandis (2013). The *P* parameter expresses the ratio of the water used for the extraction of WSOC per volume of air sampled on the filter to be analyzed. Samples were collected on Teflon (PTFE) filters in one experiment in order to estimate the particulate mass emission factor from the charbroiling of pork meat. Air was sampled from the charbroiler (just above the pork meat) at a rate of 225 L min$^{-1}$ through a custom-build exhaust line (100 mm id). A portion of these emissions were sampled through a 3/8 in line after passing through a PM$_{2.5}$ cyclone at 4 L min$^{-1}$. Known portions (25 g) of pork meat were individually cooked until well done and the emissions generated were sampled from the exhaust line.

The volatile organic compounds (VOCs) were measured by a PTR-MS (Ionicon Analytik). The drift tube was operated at 600 V at a constant pressure of 2.2-2.3 mbar. The flow rate was 0.5 L min$^{-1}$. Further information about the PTR-MS operation can be found in Kaltsonoudis et al. (2016). Blank

measurements were conducted prior to the introduction of meat cooking emissions to the chamber in each experiment. A series of gas monitors was used for the measurement of the mixing ratios of the nitrogen oxides ($NO_x$), ozone ($O_3$), carbon monoxide ($CO$), and carbon dioxide ($CO_2$) (Teledyne models: T201, 400E, 300E and T360 respectively).

Aging experiments were conducted in order to simulate the evolution of the freshly produced COA as it reacts with typical oxidants ($O_3$ and OH) in the atmosphere. UV illumination was used ($J_{NO2}$=0.59 min$^{-1}$) and the chemical evolution of the particulate and gas species was monitored. In some experiments, ozone was added and the ozonolysis of cooking emissions in the dark was investigated (Table 1). No OH precursor was used in any of the experiments.

## 2.2 Ambient measurements

Ambient aerosol was sampled at the ICE-HT institute (8 km NE from the center of Patras) during February 2012 (Kostenidou et al., 2013). This period included Fat Thursday (February 16) during which meat is charbroiled everywhere in Patras. The instrumentation used for the ambient measurements is described in Kostenidou et al. (2013). Briefly, an HR-ToF-AMS was deployed for the characterization of the non-refractory $PM_1$ aerosol composition, a PTR-MS was used for the VOCs, an SMPS for the size distributions, a MAAP for the BC, and a series of gas monitors were used for the $NO_x$, $O_3$, CO, and $CO_2$ concentrations. All instruments sampled from approximately 4 m above ground. A $PM_{2.5}$ cyclone was used in front of the MAAP.

## 2.3. Data analysis

For the HR-AMS data analysis, SQUIRREL v1.56D and PIKA v1.15D with Igor Pro 6.34A (Wavemetrics) were used, applying the fragmentation table of Aiken et al. (2008). The O:C and H:C ratios were estimated using the improved method of Canagaratna et al. (2015). High-resolution PMF analysis (Paatero and Tapper, 1994; Lanz et al., 2007) was performed using the HR-AMS data from the chamber experiments and the ambient measurements. The PMF evaluation tool PET (Ulbrich et al., 2009) was used for both cases. The Multilinear Engine (ME-2) through Source Finder software (SoFi) (Canonaco et al., 2013) was also used for the analysis of the ambient measurements to investigate the robustness of the corresponding results of the PMF. In all cases we used as inputs the m/z's 12-200 at high resolution.

The OH radical concentrations were estimated using isotopically labelled butanol (1-butanol-d9, Sigma). The change of the concentration of the PTR-MS m/z 66 was used to calculate the OH concentrations based on the second-order reaction of d9-butanol with the OH radicals. The corresponding reaction constant used is 3.4 x 10$^{12}$ cm$^3$ molecule$^{-1}$ s$^{-1}$ (Barmet et al., 2012). The wall losses corrections for the particles inside the chamber were calculated according to Pathak et al. (2007) assuming a first order loss rate for the mass concentration of the total OA. The loss rate constant was

established during the characterization period of each experiment prior to the beginning of chemical aging. The wall rate constants obtained for the experiments were in the range of 0.14 - 0.28 h$^{-1}$ and the corresponding linear fits had very high correlation coefficients. Losses of particles to the walls do remove part of the OA from the air in the chamber and make it "invisible" for our measurements. However, the observed chemical changes were relatively fast taking place mostly within a couple of hours. The corresponding time scales for losses were 4-6 hours so our conclusions are quite robust. This can be clearly seen, for example, in the dark ozonolysis experiment where fresh COA is decreasing following the O$_3$ addition significantly faster than it is lost to the walls before aging begun. However, the fact that we could not observe the corresponding potential changes to the COA particles deposited on the walls introduces some uncertainty in the results. While one would expect similar changes in these deposited particles if mass transfer of oxidants and condensable material was rapid enough, we cannot confirm this. However, the effect of wall losses of particles on the observed SOA/POA ratio is expected to be from modest to small. This has also been addressed by the work of Hildebrandt et al. (2009) who discussed the extremes of the potential fate of particles deposited on smog chamber walls.

## 3 Source characterization experiments

### 3.1 Size distribution and chemical composition of the fresh COA

Table 2 summarizes the composition of the fresh cooking aerosol for the five chamber experiments. The emitted aerosol is dominated by organic compounds (above 99 %) in all experiments. BC was on average only 0.3 % of the PM$_1$. This is consistent with the OC and EC filter analysis of the PM$_{2.5}$ aerosol that was sampled directly from the charbroiler. In these samples, the EC content for the fresh cooking emissions was less than 0.6 % of the total carbon. McDonald et al. (2003) reported that EC emissions from charbroiling and grilling of chicken and beef were 0.3 - 2.7 % of the total mass using charbroilers fuelled by natural gas. Chun-Li et al. (2015) also reported low EC emissions due to cooking in China (1.8 - 10.7% for meat roasting, 7.5% for fish roasting, 6% for snack street broiling, 1.9 % for cafeteria frying, and 10.7% for cafeteria broiling). In that study the WSOC to OC ratio was 0.05 – 0.15, indicating that the freshly emitted aerosol was mostly hydrophobic.

Figure 1a depicts the average HR-AMS mass spectra for the fresh meat charbroiling emissions. The initial spectra in all five experiments were similar with each other having angles θ of 0 to 7 degrees ($R^2$ ranging from 0.983 to 0.999). The comparison of the AMS spectra based on θ angles was favoured for the analysis of the results in the present manuscript. Briefly, a θ of 0-5 degrees shows an excellent match between the 2 spectra (with an $R^2$ ranging approximately from 1 to 0.99), a θ of 6-10 degrees shows a good match (with an $R^2$ ranging approximately from 0.98 to 0.96), a θ of 11-15 degrees shows that the two spectra have many similarities but they are not quite the same (with an $R^2$ ranging approximately from 0.95 to 0.92) and finally a θ from 16 to 30 degrees indicates spectra from different sources though there is some limited similarity ($R^2$ ranging approximately from 0.91 to 0.73). Values of

θ higher than 30 degrees suggest clearly different spectra. The advantage of the angle theta use for mass spectra comparisons is that it can represent better small differences than the $R^2$ coefficient. For example, small differences of 1-5 degrees all correspond to $R^2 = 0.99$. The initial O:C and H:C ratios based on Canagaratna et al. (2015) and based on Aiken et al. (2008) in parenthesis were on average $0.10 \pm 0.01$ ($0.08 \pm 0.01$) and $1.94 \pm 0.03$ ($1.80 \pm 0.03$) respectively. The main peaks of the corresponding AMS spectra were at m/z values 27, 29, 39, 41, 43, 55, 57, 67, 69, 71, 79, 81, 83, 91 and 95. The majority of these fragments correspond to homologous chains free of oxygen. The gas phase $CO_2$ contribution to the $CO_2^+$ signal was corrected by sampling through a HEPA filter during the experiments. The $CO_2$ levels were in the 395-435 ppm range and did not change significantly during the course of each experiment.

The number mode mobility diameter ($D_p$) of the fresh COA measured by the SMPS was 86±20 nm, while the mass mode vacuum aerodynamic diameter ($D_{va}$) measured by the AMS was 224±30 nm. Figure 2a shows the fresh COA number and mass distributions versus $D_p$ and $D_{va}$ correspondingly for Experiment 1. Figure 2b shows the mass and volume distributions versus $D_{va}$ and $D_p$ correspondingly for the fresh COA in the same experiment. The AMS and the SMPS estimated aerosol mass concentrations were quite different during all chamber experiments. The SMPS mass concentrations were lower by approximately a factor of 5 for a density of 1 g cm$^{-3}$ compared to the AMS total concentrations. Thus, an additional chamber experiment was conducted, where AMS and SMPS concentrations were compared to gravimetric measurements of the concentrations of COA samples collected on Teflon filters. For the same period the AMS mass concentration (CE=1) was 600 μg m$^{-3}$, the SMPS (assuming density 1 g cm$^{-3}$) was 100 μg m$^{-3}$, and the filter-based concentration was 500 μg m$^{-3}$. This intercomparison shows that the SMPS mass concentrations assuming spherical particles are problematic probably because the fresh particles emitted from charbroiling are non-spherical. Katrib et al. (2005) reported that during the ozonolysis of stearic acid needle-shaped particles were identified by transmission electron microscopy. SEM pictures of fresh COA particles in our experiments also suggested that the particles were not spherical. However, particles evaporate in the SEM so the proof is not conclusive. Given that CE values less than unity would further increase the disagreement between the AMS and the SMPS estimates and that the density of COA should be less than 2 g cm$^{-3}$, we estimate that the non-spherical shape of the particles introduces an error of the order of 2-4 in the volume concentration estimated by the SMPS measurements.

### 3.2. COA emission rates

Gravimetric analysis of the samples collected from above the charbroiler yielded an aerosol emission factor of 4 g kg$^{-1}$ of meat cooked. Hildemann et al. (1991) studied the emissions from hamburger cooking of regular and lean meet either by frying or charbroiling and reported emissions between 1 and 40 g kg$^{-1}$. McDonald et al. (2003) determined the emission ratios of meat cooking (hamburger, steak

and chicken) due to charbroiling or grilling and reported emission rates in the range 4 to 12 g kg$^{-1}$ (McDonald et al., 2003). These rates vary by more than one order of magnitude not only because different types of meat were cooked but also due to the different cooking procedures (charbroiling, grilling, frying etc.), and cooking specifics (well done, medium, slowly cooked, medium time, etc.).

Generally charbroiling emits more particles than frying and also the emissions increase with increasing fat content of the meat cooked. There is also additional variability related to where the meat is placed with the respect to the very hot surfaces (e.g., charcoal). In the present study we tried to duplicate the cooking conditions/practices used in Greece.

## 3.3. Emissions of volatile organic compounds

Several VOCs were emitted during cooking though their concentrations compared to the PM were low. In most cases less than 1 ppb of a specific VOC was emitted per 100 μg m$^{-3}$ of PM. The aromatic species (benzene, toluene, xylenes) were emitted in similar amounts (0.1 g kg$^{-1}$). Table 3 presents the emission factors for some of the measured VOCs based on a COA emission rate of 4 g kg$^{-1}$ of meat. To

the best of our knowledge there is little information about VOC emissions from cooking. For example, Schauer et al. (1999) has reported the emission factors from meat charbroiling over a natural gas fired grill. So our major objective was to add to this limited literature. Based on the emissions measured the SOA formation potential of cooking would be limited compared to the primary emissions. This is consistent with the limited additional SOA that we have observed experimentally as discussed in the

next section. There was no detectable decrease of the concentrations of the VOCs measured by the PTR-MS during the characterization periods.

## 3.4 Chemical aging of COA

Significant changes to the COA spectrum were observed during its oxidation. The initial and final O:C

and H:C ratios for all the experiments are reported in Table 1. For Experiment 1, in which the emissions were illuminated for 4 hours the O:C increased from 0.11 to 0.27. For Experiment 2, where the UV lights were turned on for 8 hours the O:C reached 0.30. For Experiment 3 in which dark ozonolysis took place the O:C ratio reached 0.21 two hours after the addition of 40 ppb of ozone. For experiment 4 the O:C ratio reached 0.27 after 7.5 h of exposure to UV. No change was seen for the O:C and H:C ratios

for Experiment 5 in which the COA was left in the chamber without addition of oxidants or exposure to UV.

For the experiments in which UV illumination was used the O:C ratio increased from 0.1 to 0.2 in less than 2 hours. For Experiment 3 in which 40 ppb of O$_3$ were added the O:C ratio increased from 0.12 to 0.18 in less than one hour. Figure 3a shows the temporal evolution of the O:C ratios during the

35 five smog chamber experiments. In Experiment 3 (dark ozonolysis) an increase prior to the addition of ozone was seen due to small amounts of ozone (approximately 9 ppb) initially present in the chamber.

Figure 3b presents the corresponding H:C ratio evolution during the five experiments. A reduction in H:C by 10% or so was observed in all experiments.

The theta ($\theta$) angles between the fresh and aged COA AMS spectra are summarized in Table 1. Differences in the AMS spectra between the fresh and aged COA were present throughout the m/z range. The fractional contribution of m/z 44, $f_{44}$, increased during the UV aging and the dark ozonolysis. Figure 1b shows the aged COA HR spectrum for Experiment 2 (after 8 h of UV illumination). After aging with UV for 4 h the angle $\theta$ was 22 degrees in experiment 1. The addition of ozone resulted in a 15 degree shift in 4 h. The relatively fast change in the AMS spectra is noteworthy (Figure 4). After 1 hour of UV illumination a theta angle of 12 degrees was observed. For the dark ozonolysis experiment a 15 degrees change was observed 2 hours after the ozone addition. These results indicate that the COA emitted from meat charbroiling can change rapidly after it is emitted either during the day (when it is sunny) or during the night when moderate levels of ozone are available.

Figure 5a shows the fraction of m/z 44 ($f_{44}$) and m/z 43 ($f_{43}$) as they evolve over time during Experiment 2 (8 h of UV). Most of the COA factors reported in the literature fall in the lower left part of the Ng triangle (Ng et al., 2011). After the oxidation process the system position tends to move up as $f_{44}$ increases. A similar trend was observed for the $f_{55}$ to $f_{57}$ plot (Figure 5b) where both fractions decrease due to chemical aging. A similar behaviour was observed during ozonolysis (Figures 5c and 5d).

The driving forces for these chemical aging processes were reactions with $O_3$ and OH radicals. Significant $O_3$ production was observed in the UV illumination experiments with at least 40 ppb of $O_3$ produced after a few hours of illumination. During the first hour, 15 ppb were formed and after 2 hours the ozone concentration reached 25 ppb. At the same time the OH radical concentration increased up to $5 \times 10^6$ molecules cm$^{-3}$. Figure 6a depicts the $O_3$ and OH concentrations for Experiment 2 in which UV illumination was used. Similar results were obtained for the rest of the UV illumination experiments. Figure 6b shows the $O_3$ evolution during the dark ozonolysis experiment. After the initial addition of 40 ppb of ozone, approximately 5 ppb were consumed during the first 3 hours.

Net OA production due to chemical aging was limited. In the five experiments OA mass enhancements (after corrections for particle losses) were less than 10% of the mass prior to the perturbation. This small change in mass strongly suggests that a lot of the observed chemical changes were probably due to heterogeneous reactions. Such reactions can explain the significant changes in composition (e.g., O:C) and the small additional OA formation in these experiments. Kroll et al. (2015) observed similar changes in aerosol chemical composition during exposure to OH, but at exposures that were more than one order of magnitude higher than those in our experiments. However, studies of the ozonolysis of COA components like oleic acid (e.g., Morris et al., 2002) suggest that the corresponding reactions have timescales of as little as minutes. These literature results suggest that ozonolysis was probably the dominating chemical aging pathway in our experiments.

While VOC concentrations remained stable when no UV light or oxidants were used, the concentrations of formaldehyde, acetaldehyde, formic acid, acetone, acetic acid, and methyl ethyl ketone all increased during the chemical aging. Approximately 15 ppb / 100 μg m$^{-3}$ of COA of acetaldehyde and 8 ppb / 100 μg m$^{-3}$ of COA of formaldehyde were produced during the exposure to UV and O$_3$. The increase of the concentration of these relatively small compounds suggests that fragmentation of the mostly larger organic molecules emitted during meat charbroiling is taking place. It is not clear if these molecules are products of the organics in the particulate phase (that is products of the heterogeneous reactions) or if they were produced in the gas phase

The water solubility of the COA also increased during its chemical aging. The WSOC/OC ratio for the fresh emissions was measured in each experiment and was always low with values in the 0.05 to 0.13 range. The WSOC/OC ratio after chemical aging was measured in three experiments, two after UV illumination (Experiments 1 and 2) and one after dark ozonolysis (Experiment 3). In all these three experiments the WSOC/OC ratio increased; to 0.7 for Exp. 1, 0.85 for Exp. 2, and 0.55 for Exp. 3. This shows that the WSOC to OC ratio of the aged COA is significantly higher than that of the fresh emissions and that the COA became a lot more hygroscopic as it aged.

HR-PMF analysis was performed for each chamber experiment separately. More information is provided in the Supplementary (Information SI, Section 1, Figures S1-S18). For experiments 1-4 two factors were identified: a fresh and an aged factor. Figure 7 shows the mass spectra of the two factors for Experiment 1. The mass spectra of the fresh COA factors had an O:C ratio 0.09-0.11 and they were very similar each other ($R^2>0.992$, $\theta<7^o$). They were also close to the average fresh mass spectrum from all 5 experiments ($R^2>0.96$, $\theta<10^o$). The aged factors had an O:C ratio in the range of 0.20 to 0.26 depending on the degree of oxidation. The AMS spectra of the aged factors after exposure to UV (Experiments 1, 2, and 4) were similar to each other (theta ranging from 2 to 6 degrees). The corresponding angles between the dark ozonolysis experiment (Experiment 3) and the UV exposure ones were higher ranging from 8 to 14 degrees as the dark ozonolysis factor was less oxidized. Figure 8 illustrates the time series of the 2 factors for Experiment 3 (O$_3$) and Experiment 4 (UV).

**4 Ambient measurements**

Ambient aerosol was sampled outside the ICE-HT institute (8 km NE from the city center of Patras) during February 2012 for a period of 2 days. This period included Fat Thursday (February 16) during which meat is charbroiled everywhere in Patras. The ambient measurements have been corrected for the CE, applying the algorithm of Kostenidou et al. (2007), comparing the AMS mass distributions to the SMPS volume distributions. The CE for these multi-component particles was 0.76±0.07. Applying HR PMF analysis (using PET) on the AMS spectra for these 2 days of measurements, 4 sources were identified. PMF solutions up to 5 factors were examined and evaluated, while the tested $f_{peak}$ range was between -2 and 2. More information for the selection of the factors is provided in the SI (Section 2,

Figures S19-S23). One factor was related to OOA, while the other 3 factors were attributed to primary emissions: transportation (HOA), burning of olive tree branches (otBB-OA), and meat cooking (COA). Given the small data set, the stability of the solution was further investigated using ME-2 analysis (SoFi) and applying a constrained solution for the HOA using the HOA mass spectrum of Kostenidou et al. (2013) with a=0.1 (SI, Section 3, Figures S24-S28). There was no significant change in the factors in the two solutions. More details are given in the SI (Section 3, Figures S23-S26). The cooking mass spectrum and time series did not change significant with an $R^2>0.99$ between PMF and ME-2 solutions.

Figure 9 shows the mass concentrations of the 4 factors. During midday of Fat Thursday, the organic mass concentration was 23.2 μg m$^{-3}$ representing 81% of the PM$_1$. For the same period the cooking related factor represented 85% of the organic aerosol (17.5 μg m$^{-3}$) while for the day before and the day after, the COA factor represented only the 5% of the total OA. From various studies that were conducted in Greek cities, COA appears to be 15-20% of the OA (Kostenidou et al., 2015; Florou et al., 2016). The mass spectrum of the cooking OA factor (COA) along with the rest of the factors obtained by the PMF analysis is shown in Figure 10. The m/z values contributing significantly to the COA factor were: 39, 41, 43, 44, 55, 57, 67, 69, 71 etc. which are characteristic of cooking OA found in previous studies (Ge et al., 2012; Crippa et al., 2013).

Figure 11 summarizes the angle θ between the mass spectra of the fresh and aged meat charbroiling OA and the PMF COA factors from ambient measurements from other studies. Depending on atmospheric conditions (oxidant levels) the COA AMS spectrum can be different. This can be seen by the comparison of fresh and aged COA in these experiments against the summer and winter COA factors in Greece. Other factors that appear to drive variability can include the PMF analysis itself (e.g., mixing with other sources), the type of food cooked, etc. The laboratory fresh COA spectrum has many similarities with the COA factors obtained in Athens and Patras during the winter (θ angles of 13$^o$ in both cases) especially considering that these are independent PMF results from different campaigns. On the other hand, the aged COA spectrum is similar to the cooking related factors (HOA-2) identified during the summer in both cities (θ angles of 9$^o$ for both). The spectrum of the cooking OA of Fat Thursday in Patras (a sunny period with moderate temperatures) was quite similar with the aged meat charbroiling aerosol. This demonstrates that COA ages relatively fast under ambient conditions when the necessary oxidants are available (i.e. sunny summer days) and as a result PMF analysis can distinguish only one factor (the aged COA). Even though this conversion is rapid the COA does not reach high oxidation states comparable to those of OOA. In addition, our results suggest that before performing PMF analysis using default COA spectra as external factors, one should account for their potential chemical aging.

The aged COA spectrum from our chamber experiments is similar to the COA factor reported by Sun et al. (2011) for the city of New York during summer (Figure 11). This is also true for the COA related factor (CIOA) reported by Hayes et al. (2013) for the 2010 CalNex campaign in Pasadena CA.

On the other hand, the COA factors reported for Fresno (Ge et al., 2012) and Paris (Crippa et al., 2013b) were in better agreement with the fresh COA reported in this work.

## 5 Conclusions

Particulate emissions from meat charbroiling consist mainly of organics (>99%) while BC is only 0.3%. These fresh OA emissions react rapidly with ozone and OH radicals with significant changes in their AMS spectra. After 2 hours of UV illumination (average OH concentration of 3 x $10^6$ cm$^{-3}$) the O:C ratio doubles and the aged spectrum differs approximately 15$^o$ from the fresh one. Fresh COA is hydrophobic (WSOC to OC ratio 0.05-0.15) while the aged COA is more hydrophilic (WSOC to OC ratio 0.7-0.85).

The AMS spectrum of the fresh laboratory COA was similar (theta less than 10 degrees) to the ambient PMF COA winter factors in two major Greek cities, while it was quite different (theta 20-35 degrees) than the ambient summertime COA factors. The opposite behavior was observed for the aged COA which was similar to the ambient summertime PMF COA factors (theta 10-12 degrees). These results suggest that the degree of chemical aging of the COA has to be taken into account for source identification in the PMF analysis of ambient AMS datasets.

The laboratory experiments were conducted at concentrations higher than atmospheric and this could be a potential limitation of the aging experiments. However, the concentrations in Experiment 1 when aging began were only a factor of 4 higher than the ambient COA levels shown in Figure 9, so they are by no means unreasonable. The fact that we did not observe significant differences in behaviour with initial concentration (the investigated variation was also a factor of 4) and the relatively good agreement of the aged laboratory COA AMS spectra with the ambient spectra suggests that the effect of the COA levels was probably not a serious limitation. The volatility of the cooking organic aerosol in these experiments as well as during the field campaign is discussed in a forthcoming publication.

## Acknowledgements

This research was supported by the European Research Council Project ATMOPACS (Atmospheric Organic Particulate Matter, Air Quality and Climate Change Studies) (Grant Agreement 267099) and the US Environmental Protection Agency (Grant R835873). We thank M. Elser for making the AMS spectra from Estonia available. We also thank P. L. Hayes and J. L. Jimenez for making the CIOA spectra available from CalNex campaign.

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

**Table 1.** Summary of smog chamber experiments.

| Chamber Exp.[1] | Initial PM$_1$ concentration ($\mu$g m$^{-3}$) | Aging procedure | Initial O:C | Initial H:C | Final O:C | Final H:C | Average OH (molec cm$^{-3}$) | O$_3$ formed (ppb) | θ angle initial vs final |
|---|---|---|---|---|---|---|---|---|---|
| 1 | 130 | UV illumination (4 h) | 0.11 | 1.91 | 0.27 | 1.80 | -[2] | 35 | 22 |
| 2 | 400 | UV illumination (8 h) | 0.10 | 1.91 | 0.30 | 1.76 | 2.6x10$^6$ | 47 | 27 |
| 3 | 450 | O$_3$ addition (42 ppb) | 0.10 | 1.95 | 0.21 | 1.90 | 6.5x10$^4$ | - | 16 |
| 4 | 335 | UV illumination (7.5 h) | 0.10 | 1.97 | 0.27 | 1.85 | 1.4x10$^6$ | 38 | 25 |
| 5 | 540 | None | 0.09 | 1.97 | 0.09 | 2.00 | 6.4x10$^5$ | 0 | 2.6 |

[1]The total duration of Experiments 1 to 4 was 7 to 8 hours and of Experiment 5 4.5 hours.

[2]In Experiment 1 no d-butanol was added, thus no OH radical concentration is reported.

**Table 2.** Composition (% mass) of the freshly emitted COA for the laboratory experiments.

| | Experiment | | | | | Average |
|---|---|---|---|---|---|---|
| | 1 | 2 | 3 | 4 | 5 | |
| Organics | 98.4 | 99.0 | 99.4 | 99.4 | 99.6 | 99.2 ± 0.5 |
| Sulfate | 0.1 | 0.5 | 0.1 | 0.1 | 0.1 | 0.1 ± 0.2 |
| Ammonium | 0.0 | 0.0 | 0.0 | 0.0 | 0.0 | 0.0 ± 0.0 |
| Chloride | 0.5 | 0.0 | 0.1 | 0.1 | 0.1 | 0.2 ± 0.2 |
| Nitrate | 0.2 | 0.1 | 0.2 | 0.3 | 0.2 | 0.2 ± 0.1 |
| BC | 0.8 | 0.4 | 0.2 | 0.2 | 0.1 | 0.3 ± 0.3 |

**Table 3.** Emission factors (g per Kg of meat cooked) for several VOCs.

| VOC | PTR-MS m/z | Emission rate (g kg$^{-1}$) |
|---|---|---|
| Acetonitrile | 42 | 0.01±0.00 |
| Acetone | 59 | 0.03 ±0.01 |
| Isoprene | 69 | 0.05±0.01 |
| MVK and MACR | 71 | 0.03±0.01 |
| MEK | 73 | 0.01±0.01 |
| Benzene | 79 | 0.09±0.02 |
| Toluene | 93 | 0.09±0.03 |
| Xylenes | 107 | 0.10±0.04 |
| Monoterpenes | 137 | 0.04±0.02 |

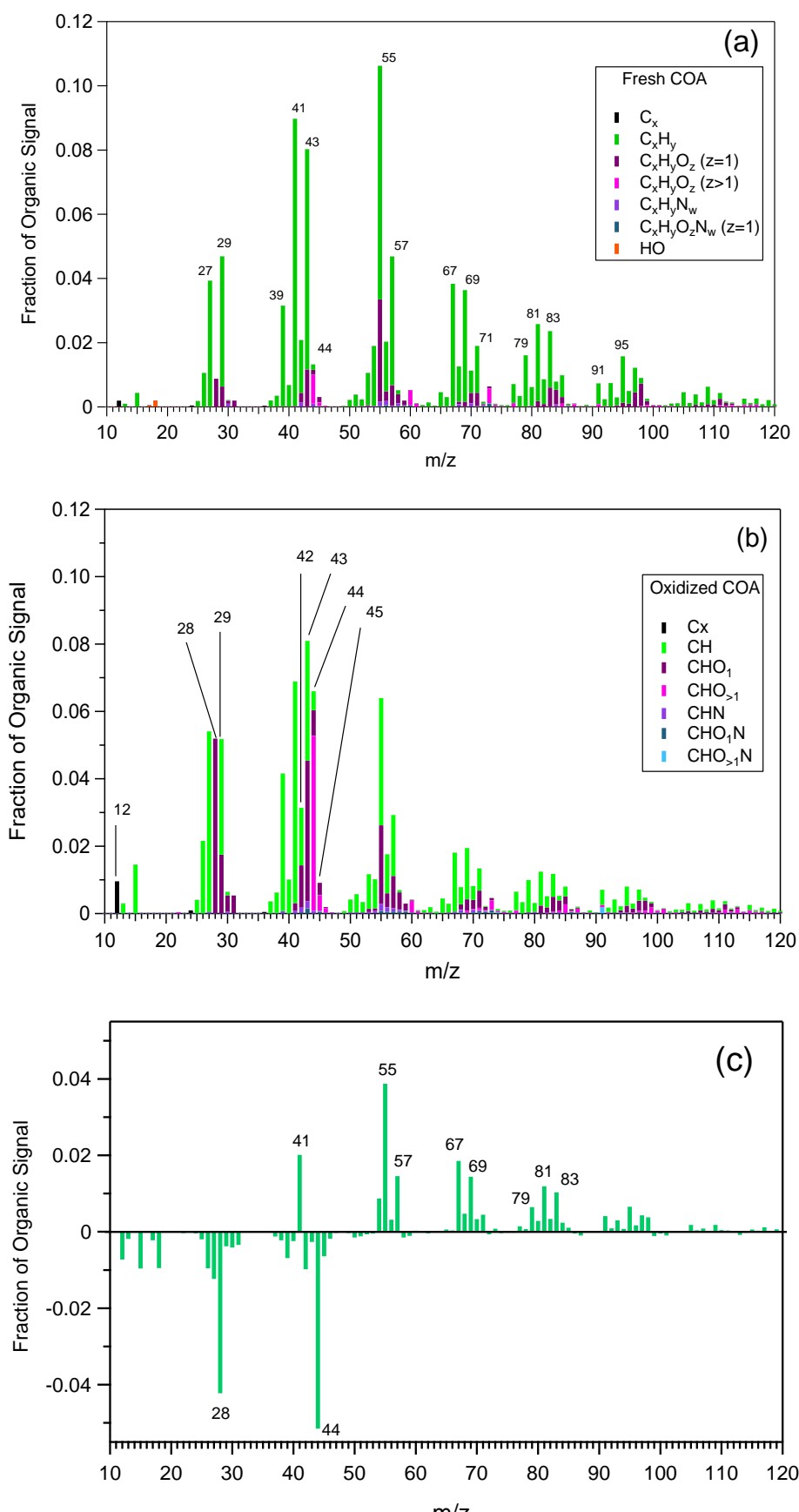

**Figure 1**. (a) Fresh COA mass spectrum, (b) aged COA mass spectrum for Experiment 2 (8 h of UV) and (c) difference between the fresh and aged COA.

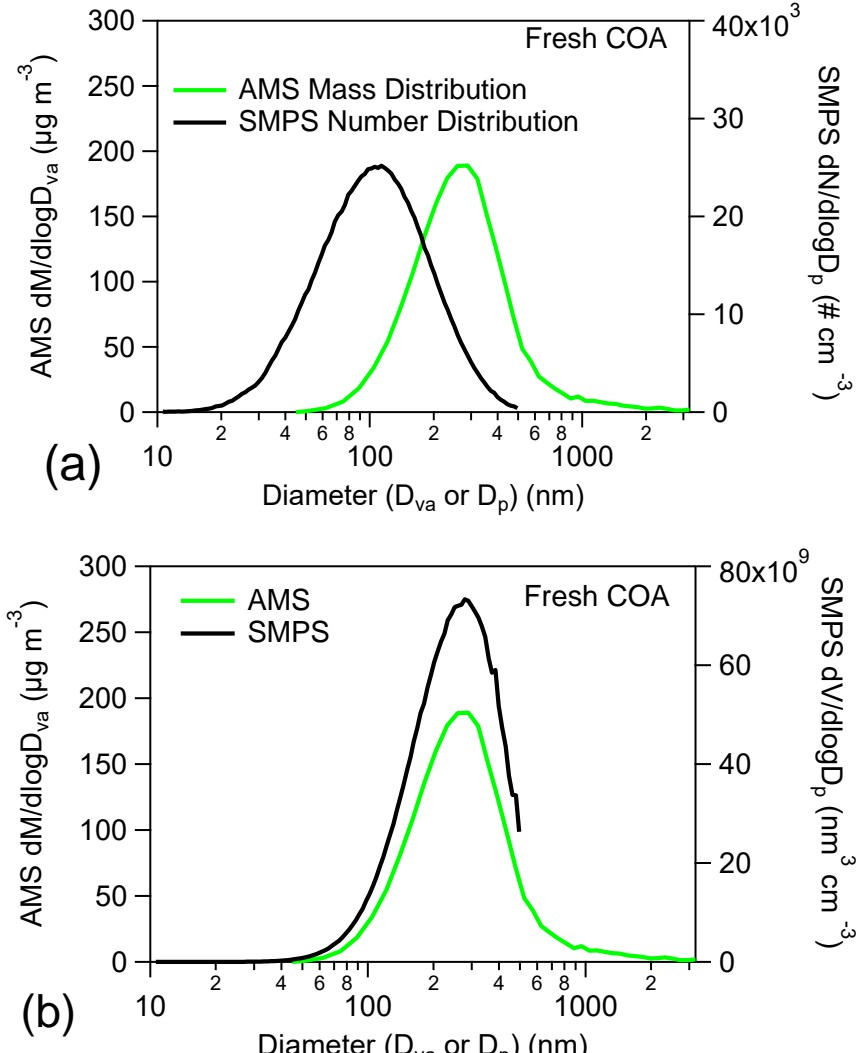

Figure 2. (a) SMPS number and AMS mass distributions versus $D_p$ and $D_{va}$ correspondingly for fresh COA and (b) SMPS volume and AMS mass distributions versus $D_{va}$ and $D_p$

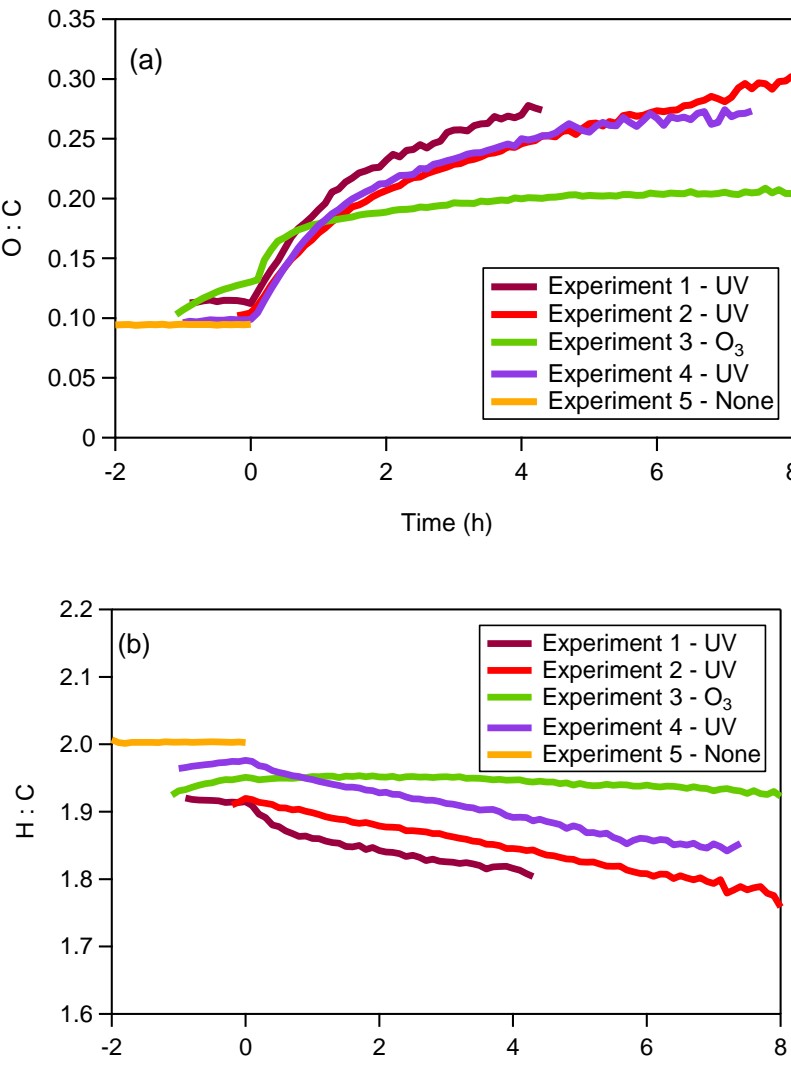

**Figure 3**. a) O:C ratios and b) H:C ratios for the COA smog chamber experiments. Time zero corresponds to the beginning of the aging process.

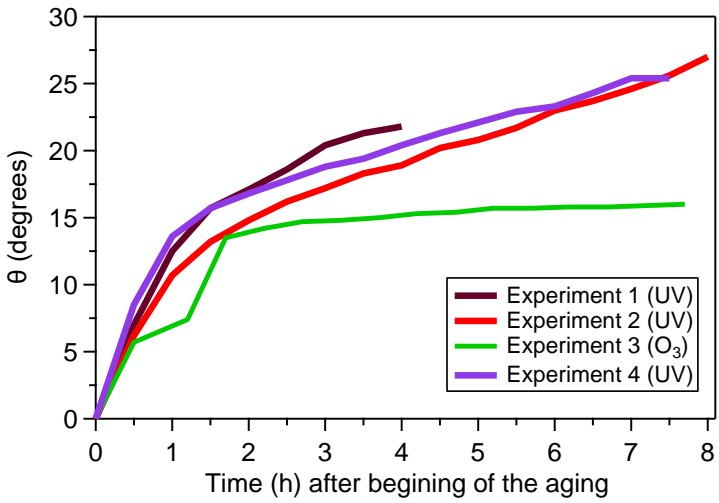

**Figure 4.** Evolution of the theta angle with the initial AMS mass spectrum during aging.

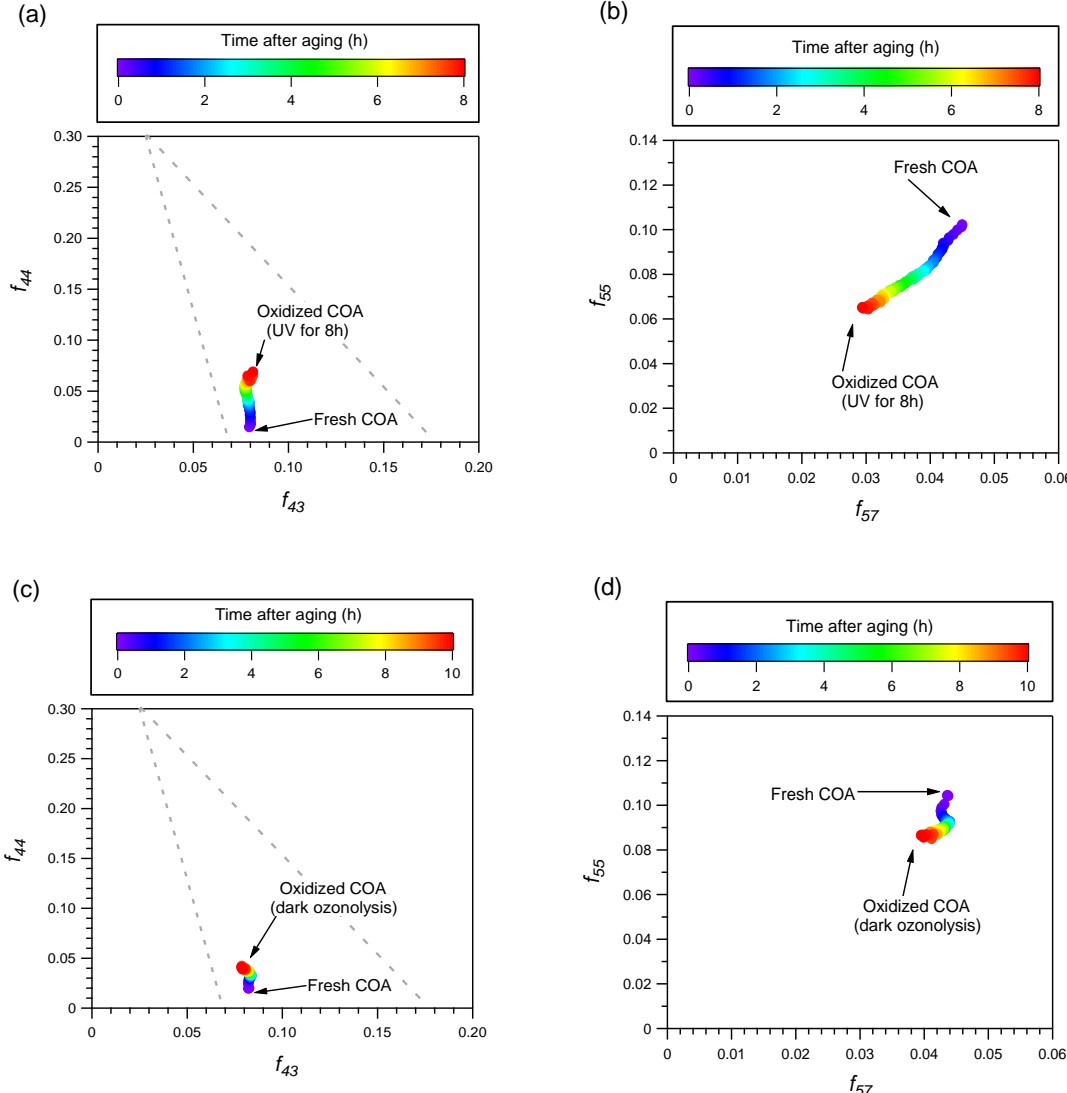

**Figure 5.** Scatter plot for the fractions of signal for Experiment 2 (UV for 8 h) and Experiment 3. (a) $f_{44}$ to $f_{43}$ for Experiment 2, (b) $f_{55}$ to $f_{57}$ for Experiment 2, (c) $f_{44}$ to $f_{43}$ for Experiment 3, and (d) $f_{55}$ to $f_{57}$ for Experiment 3.

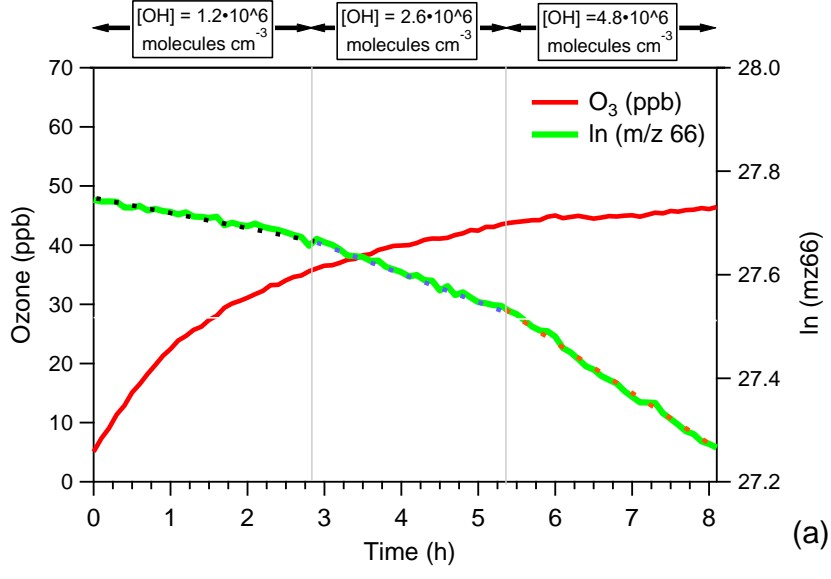

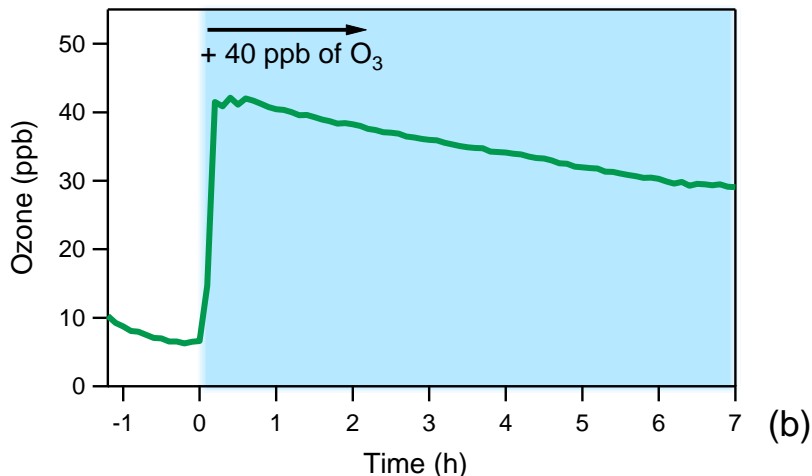

**Figure 6.** (a) Ozone and estimated OH radical (m/z 66 corresponds to d-butanol) concentrations in
5  Experiment 2 (UV illumination) and (b) Ozone concentration during Experiment 3 (dark ozonolysis).

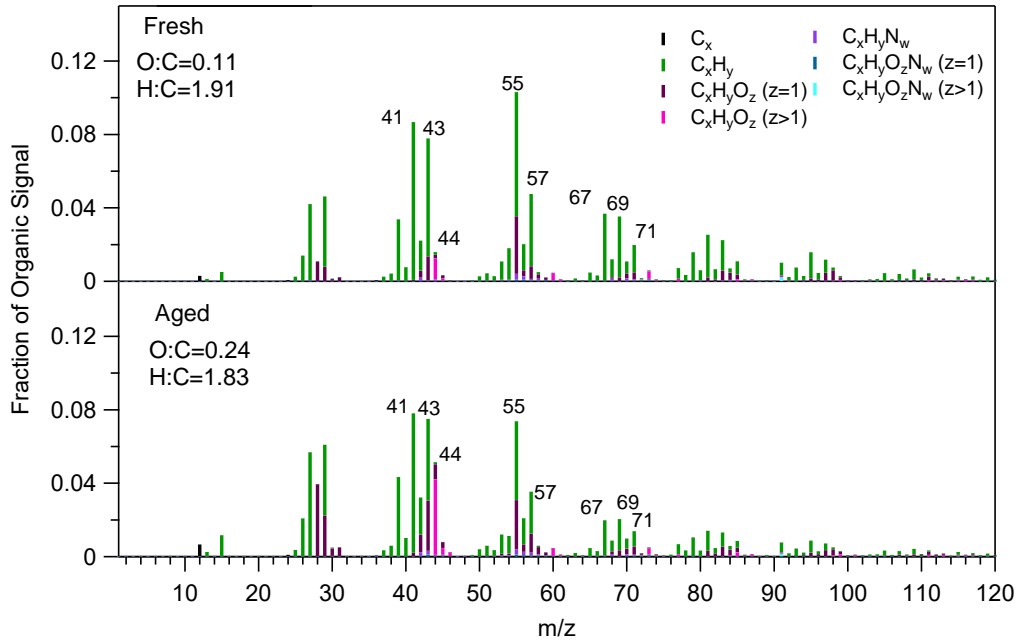

**Figure 7.** Mass spectra for the two resulting factors of the PMF analysis for Experiment 1.

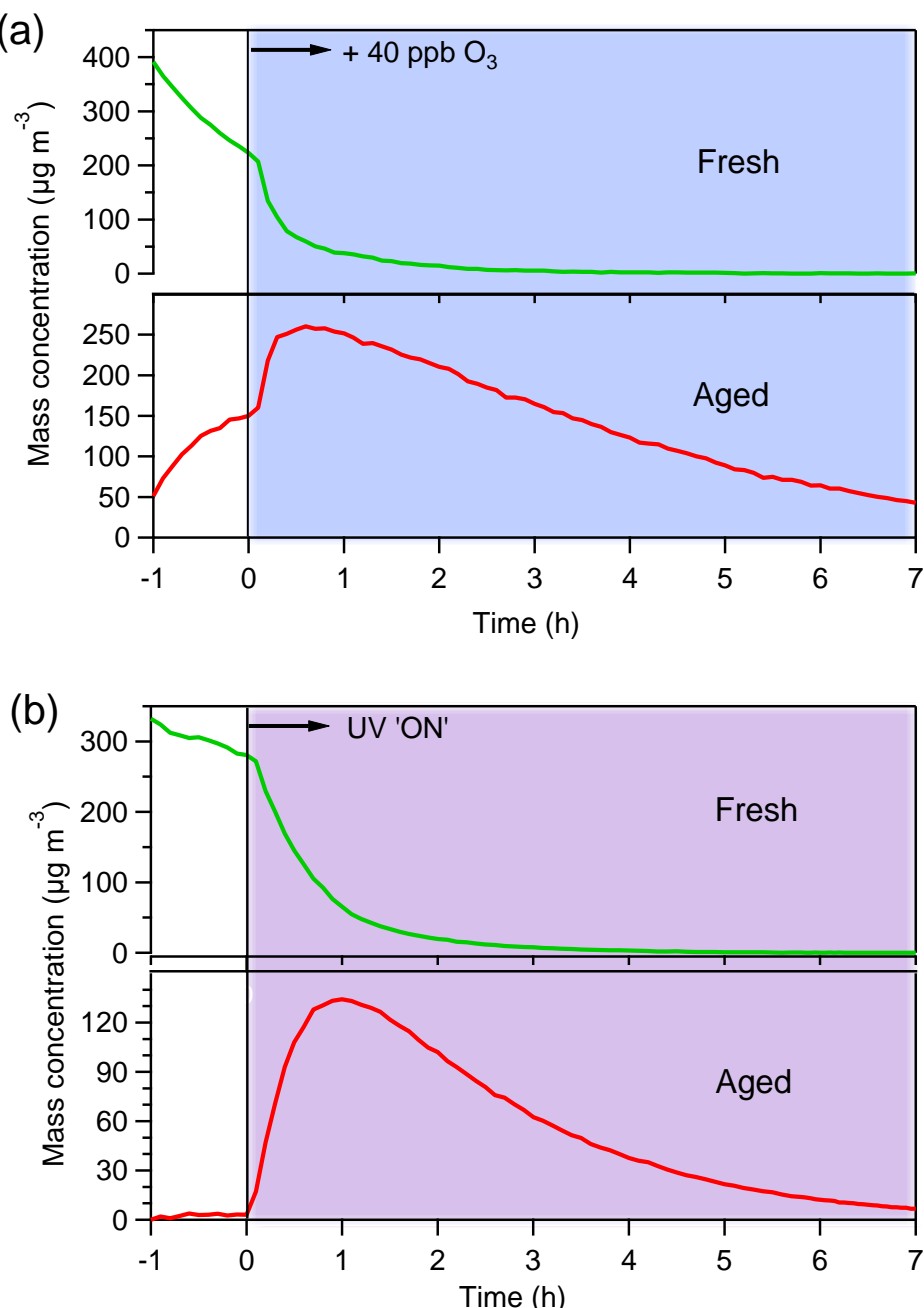

**Figure 8.** Time series of the resulting factors from the PMF analysis of the chamber experiments without corrections for losses to walls. (a) PMF factors for Experiment 3 ($O_3$ addition) and (b) PMF factors for Experiment 4 (UV illumination).

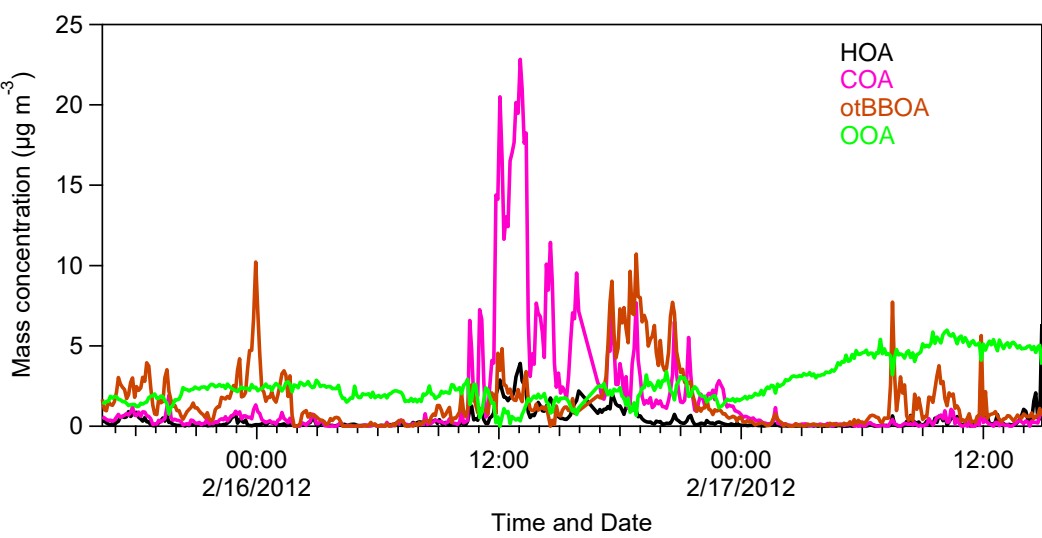

**Figure 9.** Time series of the four PMF factors found for the measurement period including Fat Thursday (16 February 2012).

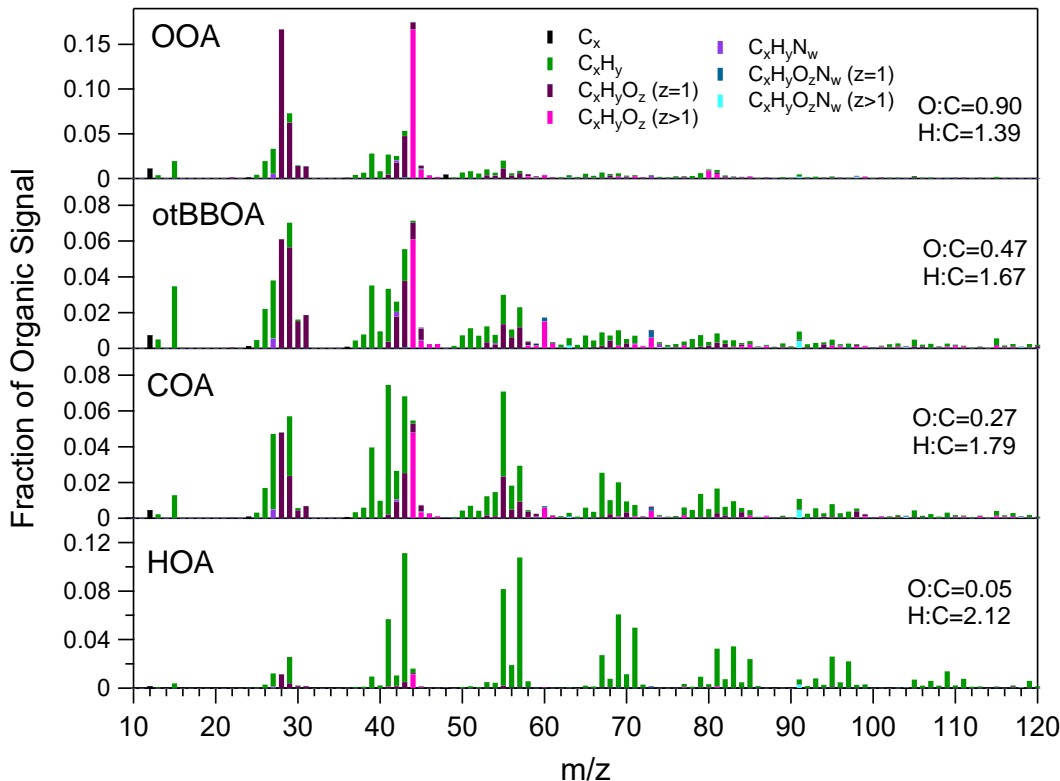

**Figure 10**. Mass spectra of the 4 PMF factors found for the measurement period including Fat Thursday (16 February 2012).

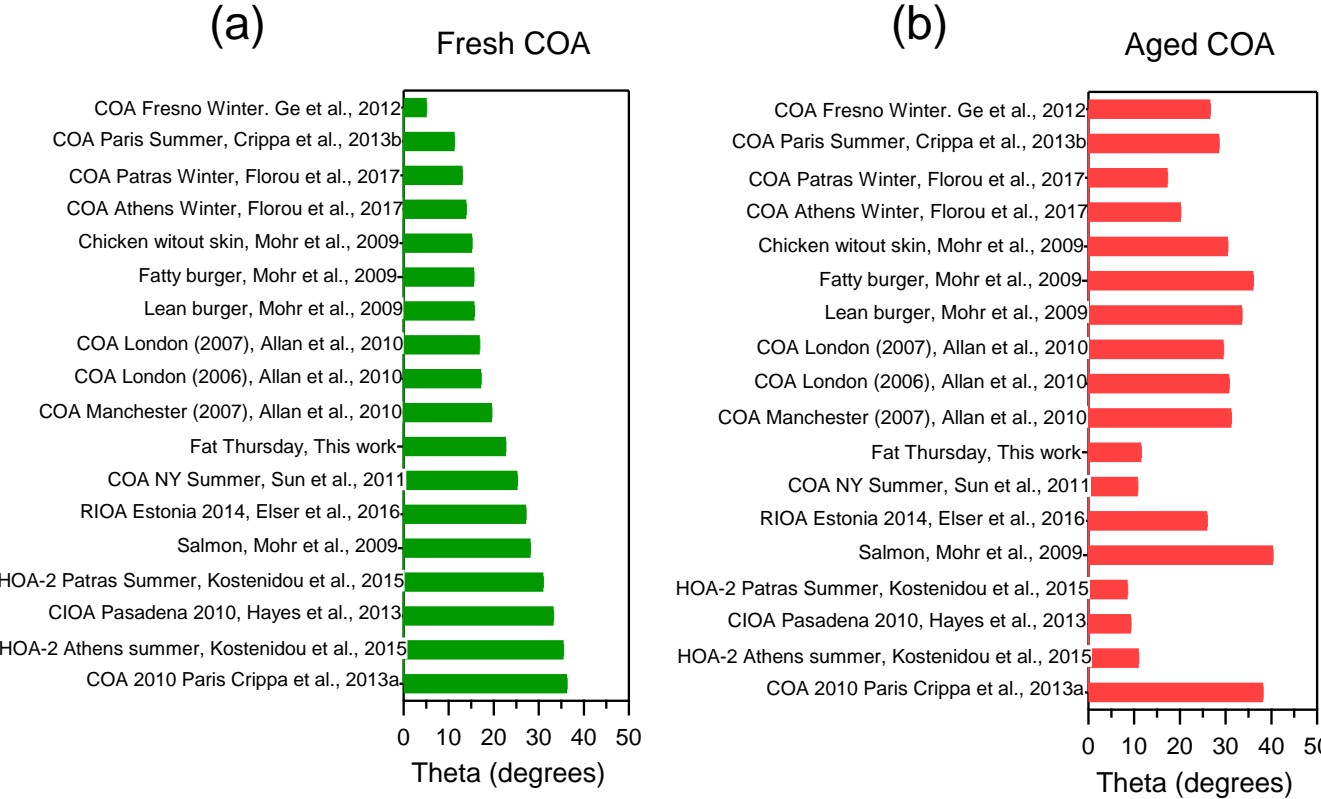

**Figure 11.** Angles θ between COA factors and the (a) fresh meat charbroiling emissions (average fresh spectrum), and (b) aged (average of the UV exposure experiments) meat charbroiling emissions.

