# Peer review of "Characterization of fresh and aged organic aerosol emissions from meat charbroiling"

_Atmospheric Chemistry and Physics, 2016_

## Referee Comment (RC1) · Anonymous Referee #1 · 21 Dec 2016

Review:

Kaltsonoudis et al. presents a series of smog chamber experiments to study the aging of primary emissions from meat charbroiling. They found that the initial and aged AMS spectra of meat charbroiling differed considerably. The derived fresh and aged cooking factors in laboratory were compared with ambient COA factors during Fat Thursday in Patras, Greece.

The experiments are novel and performed with an extensive suite of instrumentation to systematically study the chemical aging of emissions from a very important anthropogenic source, namely meat charbroiling. The paper potentially has significant implications to the formation of SOA and control of PM in many urban environments. While the datasets are interesting, the manuscript can be improved by providing more detailed interpretation of the data. The paper can be recommended for publications after the following questions are addressed.

General comments:

1. The experiments were not conducted at atmospheric relevant conditions, which could bias the conclusions and implications. As shown in Table 1, the mass concentration of $PM_1$ ranged from 130 $\mu g\ m^{-3}$ to 540 $\mu g\ m^{-3}$, much higher than typical concentrations of $PM_1$ in ambient air. The chemistry occurred at higher mass loadings of OA may be different from that at lower mass loadings. Did the authors conduct experiments at lower mass loadings of OA? If not, some discussions to relate the current findings to atmospheric implications at more realistic PM concentrations would be needed.

2. In Line 1-3, Page 6, the authors mentioned that "For the same sampling time the AMS mass concentration (CE=1) was 600 $\mu g\ m^{-3}$, the SMPS (assuming density 1 $g\ cm^{-3}$) was 100 $\mu g\ m^{-3}$, and the filter-based concentration was 500 $\mu g\ m^{-3}$." Was a CE value of 1 applied to the entire study of POA and SOA in laboratory and ambient measurements? As CE values are dependent on chemical composition of $PM_1$ (Middlebrook et al., 2012), a fixed CE value may be not suitable. In addition, gas-phase $CO_2$ will contribute to the $CO_2^+$ signal and thus influence the mass spectra (Aiken et al., 2007;Aiken et al., 2008). Was the contribution of gas-phase $CO_2$ to $CO_2^+$ signal corrected in this study? This information was missed in the manuscript. Furthermore, the authors attributed the difference of SMPS and AMS measurements to shape of the particles. It is unlikely that alone can explain the difference of 5-8 times in concentrations.

3. The authors should give a more detailed discussion of the wall loss corrections of particles. SOA formation and wall loss are competition processes that a significant wall loss would bias the measurements of the SOA in mass loading, composition, and elemental ratios etc… How would the conclusion of high POA/SOA ratio be affected

by wall loss?  In addition, the exhaust was transferred through copper tubing and a metal bellows pump (not clear if the system was heated). Did the authors characterize the losses of PM and VOC of this setup? Also, will the metal bellows pump generate particles or VOC? Any data of blank experiments with purified air?

Specific comments:

line 5, page 2: A recent study in HK suggests that COA can be 35% of OA (Lee et al., 2015)

line 19-20, page 1: "after a few hours of chemical aging" is not clear. Can the authors provide information on the OH or $O_3$ exposures?

line 6-7, page 3: What is the meaning that the cooking particles are the same? Does it mean emission rate of cooking particles?

line 17-18, page 3: A brief introduction of the chamber facility should be given.

line 24-27, page 3: What was the flow rate in the transfer line?

line 28-29, page 3: The RH and T during the experiments should be provided.

line 32-34, page 3: What is the size range of the SMPS? As suggested by the manual of SMPS 3080, the sheath flow should be set to a 10:1 ratio with the aerosol flow. Will the ratio of 5:1 set here influence the measurement of size distribution?

line 17, page 5: How was BC measured in this study? This information was missed in the manuscript.

line 27-28, page 5: The angle $\theta$ was used for the comparison between different AMS spectra throughout the entire manuscript. It would be useful to explicitly introduce the relationships between $\theta$ and spectra similarities.

Section 3.2: The authors mentioned that COA emission rates varied due to the different types of meat and cooking procedures. What are the experimental conditions for the studies of Hildemann et al. (1991) and McDonald et al. (2003)? Any suggestions on the influence of meat types and cooking procedures on COA emission rates?

Line 5, page 6: Any evidence that the particles from charbroiling are non-spherical?

Section 3.3: Emissions factors of VOCs were listed here. Any comparison or conclusions? To what extent are they related with the formation of new OA?

line 26, page 6: What was the reaction time for experiment 3?

line 32-36, page 6: It is not convincing that the increase of O:C ratios in experiment 3 was due to the initial presence of ozone. If the increase of O:C ratios are due to the reactions of particles with ozone, the concentration of ozone should have decreased prior to the addition of ozone, which is not reflected in Figure 5b. In addition, as shown in Figure 5a, similar concentration of ozone prior to the addition of ozone was also observed for experiment 2, but no increase of O:C ratios was observed for experiment 2 prior to the addition of ozone. Experiment 2 seems unique with both O:C and H:C increased prior to the addition of ozone. Are there any other explanations for this phenomenon?

line 21-22, page 7: This sentence should be mentioned prior to the description of variations of $O_3$ and OH concentration. The authors mentioned that similar results were obtained for the rest of the UV illumination experiments. Do the authors mean similar levels or trends of OH concentrations? It is suggested to provide the OH concentrations for all UV experiments.

line 25-28, page 7: What is the definition of new OA here? As the chemical composition of OA changed during aging, should the aged OA be regarded as new OA? This will largely influence the split of POA and SOA. This needs to be clarified. In addition, is there an evidence for the heterogeneous reactions?

line 29-32, page 7: The formation of carbonyls was listed. What is the implication?

line 33-35, page 7: What are the WSOC to OC ratios for the other experiments? The conclusion seems to be based on only one experiment.

line 1-8, page 8: Though detailed PMF analysis was provided in the SI, a brief introduction should be provided here. Please give some explanations on the variations of 2 factors. Did the aged COA factor show some time delay from meal hours?

Section 3.5: This section should be discussed together with the comparison of mass concentrations measured by AMS and SMPS. Also, for comparison, it is better to present the volume mode mobility diameter of particles measured by SMPS.

A table that compares the COA characteristics of this study and those reported in the literature would be useful to readers.

Technical comments:

line 3, page 3: BC and NOx are not primary organic aerosol components.

line 13, page 7: "tents" should be "tends".

Reference:

Lee, B. P., Li, Y.J., Yu, J., Louie, P., and Chan, C.K.: Characteristics of submicron particulate matter at the urban roadside in downtown Hong Kong – overview of 4 months of continuous high-resolution aerosol mass spectrometer (HR-AMS) measurements, J. Geophysical Research – Atmosphere, 10.1002/2015JD023311, 2015.

Aiken, A. C., DeCarlo, P. F., and Jimenez, J. L.: Elemental Analysis of Organic Species with Electron Ionization High-Resolution Mass Spectrometry, Analytical Chemistry, 79, 8350-8358, 10.1021/ac071150w, 2007.

Aiken, A. C., DeCarlo, P. F., Kroll, J. H., Worsnop, D. R., Huffman, J. A., Docherty, K. S., Ulbrich, I. M., Mohr, C., Kimmel, J. R., Sueper, D., Sun, Y., Zhang, Q., Trimborn, A., Northway, M., Ziemann, P. J., Canagaratna, M. R., Onasch, T. B., Alfarra, M. R., Prevot, A. S. H., Dommen, J., Duplissy, J., Metzger, A., Baltensperger, U., and Jimenez, J. L.: O/C and OM/OC Ratios of Primary, Secondary, and Ambient Organic Aerosols with High-Resolution Time-of-Flight Aerosol Mass Spectrometry, Environ Sci Technol, 42, 4478-4485, 10.1021/es703009q, 2008.

Middlebrook, A. M., Bahreini, R., Jimenez, J. L., and Canagaratna, M. R.: Evaluation of Composition-Dependent Collection Efficiencies for the Aerodyne Aerosol Mass Spectrometer using Field Data, Aerosol Sci Tech, 46, 258-271, 10.1080/02786826.2011.620041, 2012.

---

## Referee Comment (RC2) · Anonymous Referee #2 · 23 Dec 2016

Review of http://www.atmos-chem-phys-discuss.net/acp-2016-979/

Journal: Atmospheric Chemistry and Physics (ACP) Title: Characterization of fresh and aged organic aerosol emissions from meat charbroiling Author(s): Christos Kaltsonoudis, Evangelia Kostenidou, Evangelos Louvaris, Magda Psichoudaki, Epameinondas Tsiligiannis, Kalliopi Florou, Aikaterini Liangou, and Spyros N. Pandis MS No.: acp-2016-979 MS Type: Research article

The current paper reports some great novel experiments aiming to study a very important source, namely not well understood.

Cooking Organic Aerosol (COA), namely meat charbroiling. It would be good maybe to call it Meat-COA, or simply at least well state it in the abstract, where "COA" is reported but not defined.

[Figure]

As the authors state, "there are a number of remaining questions regarding the characterization of the emissions related to cooking practices." Hence, a fair description is required. The authors could do a better job in describing the available literature and recent papers on COA reported by the AMS community. I will give a number of examples that I hope can clarify and improve this great experiments carried out with an array of instruments.

- The authors do not cite the paper of Hayes, P. L., et al. (2013), Organic aerosol composition and sources in Pasadena, California during the 2010 CalNex campaign, J. Geophys. Res. Atmos., 118, 9233–9257, doi:10.1002/jgrd.50530, where it is well described a problem of COA being called Cooking Influenced Organic Aerosol (CIOA) due to the fact this factor is not uniquely associated to a single source.

- Urban increments of gaseous and aerosol pollutants and their sources using mobile aerosol mass spectrometry measurements by Elser et al 2016 (http://www.atmos-chem-phys.net/16/7117/2016/). A factor similar to COA but called Residential Influenced OA (RIOA, probably mostly from cooking processes with possible contributions from waste and coal burning), suggesting similar sources described by Dall'Osto et al (2015), issues about COA not really addressed in the current version of the paper. It is suggested to read the useful ACPD comments, may be worth to add this Elser et al study in figure 11. Taking from ACPD comments of Elser et al. (2016) "The high correlation between RIOA and published cooking mass spectra suggests that RIOA may be heavily influenced by cooking processes. However, we could not exclude the contribution from other residential sources (e.g. waste or coal combustion), especially also due to the lack of statistically robust diurnal patterns for cooking that are not affected by the drives. Therefore, we prefer to refer to this factor to RIOA, rather than cooking." Would be interesting to see what it looks like in Figure 11, and discuss briefly problems associated to COA. It is also still a pity after almost a decade of the first AMS papers related to COA, it has not been supported by external measurements.

- Model simulations of cooking organic aerosol (COA) over the UK using estimates of emissions based on measurements at two sites in London by Ri-inu Ots et al. (http://www.atmos-chem-phys.net/16/13773/2016/acp-16-13773-2016-discussion.html) discuss the fact there is potentially a factor of two in the COA AMS efficiency. It is suggested to read the ACPD comments of this paper and add in the introduction that there is still very high uncertainty on this COA AMS factor.

This is only a number of important papers stressing that "COA" is still a bit of a con-fusing factor. A better introduction and a better discussion is suggested in the major revision this paper strongly need.

Minor comment:

-pg 1 line 20, I would explain better what thetas 27° is in the text.

-pf 17. FIgure 1. I would add a part (c) with the difference between the two spectra so one can see what the positive and negative peaks are.

-Figure 9. One would argue that for the previous Wednesday and the following Friday, the emission of COA are minor. If it is important to stress 85% of OA in two hours of a spike event is important, perhaps is important to stress that the previous and following day, COA was about 5% of the OA during peak lunch and dinner times, as figure 9 suggests.

-Figure 11. It would be good to report some statistics and stress what this figure means. It looks that the difference of the Thetas are only in Sun 2011 and Ge 2010. It would be useful to add other factors partially due to cooking and see if they match more or less (it would be good to add the factors of Elser 2016 and Hayer 2013, showing they do not match with the current pork meat cooking COA herein reported).

---

## Author Comment (AC1) · 6 Mar 2017

*Kaltsonoudis et al. presents a series of smog chamber experiments to study the aging of primary emissions from meat charbroiling. They found that the initial and aged AMS spectra of meat charbroiling differed considerably. The derived fresh and aged cooking factors in laboratory were compared with ambient COA factors during Fat Thursday in Patras, Greece. The experiments are novel and performed with an extensive suite of instrumentation to systematically study the chemical aging of emissions from a very important anthropogenic source, namely meat charbroiling.*

*The paper potentially has significant implications to the formation of SOA and control of PM in many urban environments. While the datasets are interesting, the manuscript can be improved by providing more detailed interpretation of the data. The paper can*

[Figure]

*be recommended for publications after the following questions are addressed.*

*General comments:*

**(1)** *The experiments were not conducted at atmospheric relevant conditions, which could bias the conclusions and implications. As shown in Table 1, the mass concentration of PM1 ranged from 130 to 540 micrograms per cubic meter, much higher than typical concentrations of PM1 in ambient air. The chemistry occurred at higher mass loadings of OA may be different from that at lower mass loadings. Did the authors conduct experiments at lower mass loadings of OA? If not, some discussions to relate the current findings to atmospheric implications at more realistic PM concentrations would be needed.*

This is a good point. The experiments were conducted at concentrations higher than atmospheric and this could be a potential limitation of the aging experiments. However, the concentration in Experiment 1 when aging began was only a factor of 4 higher than the ambient COA levels shown in Figure 9, so these levels are by no means unreasonable at least initially. The fact that we did not observed significant differences in behavior with initial concentration (the investigated variation was also a factor of 4) and the relatively good agreement of the aged laboratory COA AMS spectra with the ambient spectra suggests that the effect of the COA levels was probably not a serious problem. This point and the corresponding limitation of the present work is now discussed in the end of the Conclusions section of the revised paper.

**(2)** *In Line 1-3, Page 6, the authors mentioned that "For the same sampling time the AMS mass concentration (CE=1) was 600 $\mu g\ m^{-3}$, the SMPS (assuming density 1 g $cm^{-3}$) was 100 $\mu g\ m^{-3}$, and the filter-based concentration was 500 $\mu g\ m^{-3}$." Was a CE value of 1 applied to the entire study of POA and SOA in laboratory and ambient measurements? As CE values are dependent on chemical composition of PM1 (Middlebrook et al., 2012), a fixed CE value may be not suitable. In addition, gas-phase*

CO2 will contribute to the $CO_2^+$ signal and thus influence the mass spectra (Aiken et al., 2007; Aiken et al., 2008). *Was the contribution of gas-phase CO2 to $CO_2^+$ signal corrected in this study? This information was missed in the manuscript. Furthermore, the authors attributed the difference of SMPS and AMS measurements to shape of the particles. It is unlikely that alone can explain the difference of 5-8 times in concentrations.*

A CE=1 was applied for both fresh and aged OA for the laboratory experiments. The composition-dependent collection efficiency of Middlebrook et al. (2012) is based on the mass fractions of ammonium, nitrate and sulfate in the total $PM_1$. In our case, the COA was almost entirely composed of organics, thus the above method is not directly applicable to our laboratory results. The ambient measurements have been corrected for the CE, applying the algorithm of Kostenidou et al. (2007), comparing the AMS mass distributions to the SMPS volume distributions. The CE for these multicomponent particles was 0.76±0.07. This information has been added to the corresponding section.

We have corrected for the gas phase $CO_2$ contribution to the $CO_2^+$ signal by sampling through a HEPA filter during the experiments. The $CO_2$ levels were in the 395-435 ppm range and did not change significantly during the course of each experiment. This information has been added in the revised manuscript.

We agree that the shape of the particles is only one factor that leads to the apparent discrepancies between these SMPS and AMS-based OA concentrations. Other factors include the CE, the particle density, the different size ranges of the two instruments, etc. This is now explained in the text.

**(3)** *The authors should give a more detailed discussion of the wall loss corrections of particles. SOA formation and wall loss are competition processes that a significant wall loss would bias the measurements of the SOA in mass loading, composition, and elemental ratios etc. How would the conclusion of high POA/SOA ratio be affected*

*by wall loss? In addition, the exhaust was transferred through copper tubing and a metal bellows pump (not clear if the system was heated). Did the authors characterize the losses of PM and VOC of this setup? Also, will the metal bellows pump generate particles or VOC? Any data of blank experiments with purified air?*

Wall loses in the chamber were calculated assuming a first order loss rate for the mass concentration of the total OA. The loss rate constant was established during the characterization period of each experiment prior to the beginning of chemical aging. The wall rate constants obtained for the experiments were in the range of 0.14 - 0.28 $h^{-1}$ and the corresponding linear fits had very high correlation coefficients.

Losses of particles to the walls do remove part of the OA from the air in the chamber and make it "invisible" for our measurements. However, the observed chemical changes were relatively fast taking place mostly within a couple of hours. The corresponding time scales for losses were 4-6 hours so our conclusions are quite robust. This can be clearly seen, for example, in the dark ozonolysis experiment where fresh COA is decreasing following the ozone addition significantly faster than it is lost to the walls before aging begun (see Figure 7a). However, the fact that we could not observe the corresponding potential changes to the COA particles deposited on the walls introduces some uncertainty in the results. While one would expect similar changes in these deposited particles if mass transfer of oxidants and condensable material was rapid enough, we cannot confirm this. However, the effect of wall losses of particles on the observed SOA/POA ratio is expected to be small modest to small. A summary of this discussion with a reference to the work of Hildebrandt et al. (2009) who discussed the extremes of the potential fate of particles deposited on smog chamber walls has been added to the revised paper.

The copper tubing used for the sampling was insulated and was therefore heated by the exhaust vapors. Its length was less than 2 m. We have confirmed that the metal bellows pump, as expected based on its design, does not generate particles or VOCs. The $PM_1$ losses in the Metal Belows pump have been characterized previously (Kostenidou et

al. 2013) using 2 SMPS systems for both ammonium sulfate and ambient particles. The losses were less than 10 percent for particles larger than 150 nm, increasing to 30 percent for 100 nm particles. This additional information has been added to the manuscript.

*Specific comments:*

**(4)** *Line 5, page 2: A recent study in HK suggests that COA can be 35 percent of OA (Lee et al., 2015).*

The proposed reference has been added to the revised manuscript.

**(5)** *Line 19-20, page 1: "after a few hours of chemical aging" is not clear. Can the authors provide information on the OH or ozone exposures?*

We now clarify that the corresponding exposures were of the order of $10^{10}$ molecules $cm^{-3}$ s for OH and 100 ppb hr for ozone.

**(6)** *Line 6-7, page 3: What is the meaning that the cooking particles are the same? Does it mean emission rate of cooking particles?*

This sentence refers to the cooking practices in Greece during winter and summer. Due to the mild climate there is no significant change in what is cooked during the different seasons (as opposed for example to cities in much colder climates). We have rephrased the sentence to avoid confusion.

**(7)** *Line 17-18, page 3: A brief introduction of the chamber facility should be given.*

A short description of the chamber facility has been added in the revised manuscript.

**(8)** *Line 24-27, page 3: What was the flow rate in the transfer line? Line 28-29, page 3: The RH and T during the experiments should be provided.*

The flow rate for the transfer line was approximately 170 L min$^{-1}$. The T and RH were in the range of 20-25$^o$C and 15-35 percent respectively. The above information has been added to the revised manuscript.

**(9)** *Line 32-34, page 3: What is the size range of the SMPS? As suggested by the manual of SMPS 3080, the sheath flow should be set to a 10:1 ratio with the aerosol flow. Will the ratio of 5:1 set here influence the measurement of size distribution?*

The size range of the SMPS under this configuration is from 10 to 500 nm. The 10:1 ratio provides more accurate size distribution measurements as the instrument has a sharper transfer function but it reduces the measurement range to 10-300 nm. Given the modest size accuracy requirements in this study (a few percent), we selected to cover a larger size range instead.

**(10)** *Line 17, page 5: How was BC measured in this study? This information was missed in the manuscript.*

A Multiple-Angle Absorption Photometer (MAAP, Thermo Scientific Inc.) with a PM$_1$ cyclone was used for the BC measurements (please see lines 34-35 in page 3 of the original manuscript).

**(11)** *Line 27-28, page 5: The angle $\theta$ was used for the comparison between different AMS spectra throughout the entire manuscript. It would be useful to explicitly introduce the relationships between $\theta$ and spectra similarities.*

A short discussion of the various similarity measures (e.g., the coefficient of determination) and the angle $\theta$ has been added to the revised manuscript.

**(12)** *Section 3.2: The authors mentioned that COA emission rates varied due to the different types of meat and cooking procedures. What are the experimental conditions*

*for the studies of Hildemann et al. (1991) and McDonald et al. (2003)? Any suggestions on the influence of meat types and cooking procedures on COA emission rates?*

Hildemann et al. (1991) studied the emissions from hamburger cooking of regular and lean meet either by frying or charbroiling. McDonald et al. (2003) determined the emission ratios of meat cooking (hamburger, steak and chicken) due to charbroiling or grilling. Generally charbroiling emits more particles than frying and also the emissions increase with increasing fat content of the meat cooked. There is also additional variability related to where the meat is placed with the respect to the very hot surfaces (e.g., charcoal). In the present study we tried to duplicate the cooking conditions/practices used in Greece.

**(13)** *Line 5, page 6: Any evidence that the particles from charbroiling are non-spherical?*

We have SEM pictures of fresh COA particles that suggest that the particles are not spherical. However, particles evaporate in the SEM so the proof is not conclusive. The second piece of evidence is the disagreement between the SMPS and AMS measurements, while the AMS results are consistent with the filter measurements.

**(14)** *Section 3.3: Emissions factors of VOCs were listed here. Any comparison or conclusions? To what extent are they related with the formation of new OA?*

To the best of our knowledge there is little information about VOC emissions from cooking. For example, Schauer et al. (1999) has reported the emission factors from meat charbroiling over a natural gas fired grill. So our major objective was to add to this limited literature. Based on the emissions measured the SOA formation potential of cooking would be limited compared to the primary emissions. This is consistent with the limited additional SOA that we have observed experimentally. This discussion has been added to the revised paper.

**(15)** *Line 26, page 6: What was the reaction time for experiment 3?*

It took two hours for the O:C ratio to reach 0.21 in this experiment. The experiment lasted 7 hours after the ozone addition. This is now mentioned in the paper.

**(16)** *Line 32-36, page 6: It is not convincing that the increase of O:C ratios in experiment 3 was due to the initial presence of ozone. If the increase of O:C ratios are due to the reactions of particles with ozone, the concentration of ozone should have decreased prior to the addition of ozone, which is not reflected in Figure 5b. In addition, as shown in Figure 5a, similar concentration of ozone prior to the addition of ozone was also observed for experiment 2, but no increase of O:C ratios was observed for experiment 2 prior to the addition of ozone. Experiment 2 seems unique with both O:C and H:C increased prior to the addition of ozone. Are there any other explanations for this phenomenon?*

This is an interesting observation. We have updated Figure 5b in the paper to show better what happened before the beginning of the aging phase. Ozone concentrations decreased from approximately 10 ppb to 6 ppb in this phase within one hour so this is consistent with our hypothesis. In experiment 2 in which the initial ozone was 5 ppb, there was little change during the characterization phase. Of course, there are other differences in Experiment 3 (e.g., the highest initial OA concentration) that could have played a role in the results.

**(17)** *Line 21-22, page 7: This sentence should be mentioned prior to the description of variations of ozone and OH concentration. The authors mentioned that similar results were obtained for the rest of the UV illumination experiments. Do the authors mean similar levels or trends of OH concentrations? It is suggested to provide the OH concentrations for all UV experiments.*

The OH and ozone concentrations for Experiments 2, 4, and 5 in which UV illumination was used were quite similar. Table 1 has been updated to include the average OH

radical concentration for the chamber experiments. For Experiment 1 the OH radical concentration was not measured (no d-butanol was added).

**(18)** *Line 25-28, page 7: What is the definition of new OA here? As the chemical composition of OA changed during aging, should the aged OA be regarded as new OA? This will largely influence the split of POA and SOA. This needs to be clarified. In addition, is there an evidence for the heterogeneous reactions?*

Given that there are both homogeneous and heterogeneous reactions taking place in the system we have rephrased this deleting the characterization "new" referring instead to "net OA production". Also given the complexity of the situation we have avoided the use of the term SOA, as based on the traditional definition SOA formation requires gas-to-particle conversion. The evidence for heterogeneous reaction is indirect and is based on the significant changes in composition (e.g., O:C) that cannot be explained by the small additional OA formation in these experiments. This important point is clarified in the revised paper.

**(19)** *Line 29-32, page 7: The formation of carbonyls was listed. What is the implication?*

The increase of the concentration of these relatively small compounds suggests the fragmentation of the mostly larger organic molecules emitted during meat charbroiling. It is not clear if these molecules are products of the organics in the particulate phase (that is products of the heterogeneous reactions) or if they were produced in the gas phase. This point has been added to the paper.

**(20)** *Line 33-35, page 7: What are the WSOC to OC ratios for the other experiments? The conclusion seems to be based on only one experiment.*

The WSOC/OC ratio for the fresh emissions was measured in each experiment and was always low with values in the 0.05 to 0.13 range. The WSOC/OC ratio was measured in three experiments, two after UV illumination (Experiments 1 and 2) and one

after dark ozonolysis (Experiment 3). In all these three experiments the WSOC/OC ratio increased dramatically to 0.7 for Exp. 1, 0.85 for Exp. 2, and 0.55 for Exp. 3. This information has been added to the paper.

**(21)** *Line 1-8, page 8: Though detailed PMF analysis was provided in the SI, a brief introduction should be provided here. Please give some explanations on the variations of 2 factors. Did the aged COA factor show some time delay from meal hours?*

A brief summary of the PMF analysis has been added. The two COA factors from the PMF analysis of the chamber experiments were quite similar to the spectra obtained from the PMF analysis of ambient air. Given that the preparation of the food often starts before meal hours and atmospheric dispersion mixes the emissions from different parts of the city, it was difficult to conclude something about any potential time delay.

**(22)** *Section 3.5: This section should be discussed together with the comparison of mass concentrations measured by AMS and SMPS. Also, for comparison, it is better to present the volume mode mobility diameter of particles measured by SMPS. A table that compares the COA characteristics of this study and those reported in the literature would be useful to readers.*

The material of this section has been moved to Section 3.1 which is now called "Size distribution and chemical composition of the fresh COA". We have added the information about the volume mode mobility diameter of the particles as measured by the SMPS. We would prefer not to add the recommended table because of the many differences of the various studies.

*Technical comments:*

**(23)** *Line 3, page 3: BC and NOx are not primary organic aerosol components.*

We have rephrased this sentence.

[Figure]

**(24)** *Line 13, page 7: "tents" should be "tends".*

We have corrected the typo.

**References:**

Lee, B.P., Li, Y.J.,Yu, J., Louie, P., and Chan, C.K.: Characteristics of submicron particulate matter at the urban roadside in downtown Hong Kong – overview of 4 months of continuous high-resolution aerosol mass spectrometer (HR-AMS) measurements, J. Geophys. Res., 10.1002/2015JD023311, 2015.

Aiken, A. C., DeCarlo, P. F., and Jimenez, J. L.: Elemental analysis of organic species with Electron Ionization High-Resolution Mass Spectrometry, Analytical Chemistry, 79, 8350-8358,10.1021/ac071150w, 2007.

Aiken, A. C., DeCarlo, P. F., Kroll, J. H., Worsnop, D. R., Huffman, J. A., Docherty, K. S., Ulbrich, I. M., Mohr, C., Kimmel, J. R., Sueper, D., Sun, Y., Zhang, Q., Trimborn, A., Northway, M., Ziemann, P. J., Canagaratna, M. R., Onasch, T. B., Alfarra, M. R., Prevot, A. S. H., Dommen, J., Duplissy, J., Metzger, A., Baltensperger, U., and Jimenez, J. L.: O/C and OM/OC Ratios of primary, secondary, and ambient organic aerosols with High-Resolution Time-of-Flight Aerosol Mass Spectrometry, Environ. Sci. Technol., 42, 4478-4485, 10.1021/es703009q, 2008.

Middlebrook, A. M., Bahreini, R., Jimenez, J. L., and Canagaratna, M. R.: Evaluation of composition-dependent collection efficiencies for the Aerodyne Aerosol Mass Spectrometer using field data, Aerosol Sci. Tech., 46, 258-271, 10.1080/02786826.2011.620041, 2012.

Schauer, J.J., Kleeman, M.J., Cass, G., and Simoneit, B.T.: Measurement of emissions from air pollution sources. 1. C1 through C29 organic compounds from meat charbroiling, Environ. Sci. Technol., 33, 1566-1577, 1999.

---

## Author Comment (AC2) · 6 Mar 2017

**(1)** *The current paper reports some great novel experiments aiming to study a very important source, namely not well understood. Cooking Organic Aerosol (COA), namely meat charbroiling. It would be good maybe to call it Meat-COA, or simply at least well state it in the abstract, where "COA" is reported but not defined.*

We appreciate the positive assessment of our work. We now clarify in the abstract that we are referring to meat charbroiling. We would prefer to keep the term COA in the rest of the paper for simplicity.

**(2)** *As the authors state, "there are a number of remaining questions regarding the characterization of the emissions related to cooking practices." Hence, a fair descrip-*

*tion is required. The authors could do a better job in describing the available literature and recent papers on COA reported by the AMS community. I will give a number of examples that I hope can clarify and improve this great experiments carried out with an array of instruments.*

*The authors do not cite the paper of Hayes, P. L., et al. (2013), Organic aerosol composition and sources in Pasadena, California during the 2010 CalNex campaign, J. Geophys. Res. Atmos., 118, 9233–9257, doi:10.1002/jgrd.50530, where it is well described a problem of COA being called Cooking Influenced Organic Aerosol (CIOA) due to the fact this factor is not uniquely associated to a single source. Urban increments of gaseous and aerosol pollutants and their sources using mobile aerosol mass spectrometry measurements by Elser et al 2016 (http://www.atmos-chem-phys.net/16/7117/2016/). A factor similar to COA but called Residential Influenced OA (RIOA, probably mostly from cooking processes with possible contributions from waste and coal burning), suggesting similar sources described by Dall Osto et al (2015), issues about COA not really addressed in the current version of the paper. It is suggested to read the useful ACPD comments, may be worth to add this Elser et al study in figure 11. Taking from ACPD comments of Elser et al. (2016) "The high correlation between RIOA and published cooking mass spectra suggests that RIOA may be heavily influenced by cooking processes. However, we could not exclude the contribution from other residential sources (e.g. waste or coal combustion), especially also due to the lack of statistically robust diurnal patterns for cooking that are not affected by the drives. Therefore, we prefer to refer to this factor to RIOA, rather than cooking."Would be interesting to see what it looks like in Figure 11, and discuss briefly problems associated to COA. It is also still a pity after almost a decade of the first AMS papers related to COA, it has not been supported by external measurements. Model simulations of cooking organic aerosol (COA) over the UK using estimates of emissions based on measurements at two sites in London by Riinu Ots et al. (http://www.atmos-chem-phys.net/16/13773/2016/acp-16-13773-2016-discussion.html) discuss the fact there is potentially a factor of two in the COA AMS efficiency. It is suggested to read the ACPD*

*comments of this paper and add in the introduction that there is still very high uncertainty on this COA AMS factor. This is only a number of important papers stressing that "COA" is still a bit of a confusing factor. A better introduction and a better discussion is suggested in the major revision this paper strongly need.*

We agree with the point of the reviewer that a lot of the controversy regarding COA has resulted from ambient AMS studies. We have followed the corresponding suggestion and improved the introduction and the corresponding discussion in the revised manuscript.

*Minor comments:*

**(2)** *Page 1 line 20, I would explain better what thetas 27 degrees is in the text.*

A brief explanation has been added.

**(3)** *Page 17. Figure 1. I would add a part (c) with the difference between the two spectra so one can see what the positive and negative peaks are.*

Figure 1 has been updated to include the suggested difference of the spectra.

**(4)** *Figure 9. One would argue that for the previous Wednesday and the following Friday, the emission of COA are minor. If it is important to stress 85 percent of OA in two hours of a spike event is important, perhaps is important to stress that the previous and following day, COA was about 5 percent of the OA during peak lunch and dinner times, as Figure 9 suggests.*

The proposed comment has been included in the revised manuscript. On average COA appear to be 15-20 percent of the OA in major Greek cities.

**(5)** *Figure 11. It would be good to report some statistics and stress what this figure means. It looks that the difference of the Thetas are only in Sun 2011 and Ge 2010. It*

*would be useful to add other factors partially due to cooking and see if they match more or less (it would be good to add the factors of Elser 2016 and Hayer 2013, showing they do not match with the current pork meat cooking COA herein reported).*

This is a good point. Figure 11 has been updated and it now includes comparisons to additional studies. One of the points of this figure is that depending on atmospheric conditions (oxidant levels) the COA AMS spectrum can be different. This can be seen by the comparison of fresh and aged COA in these experiments against the summer and winter COA factors in Greece. Other factors that appear to drive variability can include the PMF analysis itself (e.g., mixing with other sources), the type of food cooked, etc. This discussion has been added to the paper.

---

## Editor Decision (ED1)

**Comments:**

I would like to see a more thorough discussion on how the chemistry in the chamber experiments would be expected to be similar to or different from that occurring in the atmosphere. In particular, I would like to see evaporation of POA, loss of gas-phase species to walls, $NO_x$ concentration, and $NO/NO_2$ partitioning to be addressed.

Page 7 line 22: A change in the CE would presumably make the AMS vs. SMPS comparison worse. Please revise and expand this section to provide information on how the suggested factors would influence the comparison and what order of magnitude effect would be expected.

Page 9 lines 21-23: There is a significant body of literature regarding heterogeneous oxidation reactions (e.g., Kroll et al. (2015) and references therein). Are the results presented here consistent with previous works in terms of the O:C and H:C changes observed at the levels of oxidant exposure achieved in these experiments? Given the large body of work on heterogeneous oxidation, a more thorough discussion of this here would be prudent.

Page 10 lines 8-10: How different in terms of theta were the aged factors from each other? For the aged factor used later in the paper (particularly in Figure 11), was an average aged factor used, or one from a specific experiment? Was there a noticeable difference in between the ozonolysis only factor and those aged with OH?

Page 11 line 7: Here a theta of 13 degrees is defined as "quite similar" whereas on page 6 line 28 a theta of 11-15 is discussed as having "many similarities though they are not the same" and in Sect. 3.4 a change of 15 degrees is discussed as being significant in terms of the changes observed due to ozonolysis. Likewise, the use of "significantly" on page 11 line 27 in describing a change of 15 degrees should be reconsidered. While I recognize that this analysis is somewhat qualitative, it would be beneficial to maintain more consistent descriptions throughout the manuscript.

**Technical Corrections:**

Page 2 line 32: Suggest change to "..indicate that commercial and residential cooking contribute to…"

Page 3 line 3: Suggest change to "…may significantly alter…"

Page 3 line 7: BC is not a primary organic aerosol component.

Page 3 line 23-24: What is the size of the chamber itself?

Section 2.1: Please clarify if an OH precursor was used.

Page 4 line 8: The voltage difference between the filament and the ion chamber is 70 V.

Page 4 line 21: "P parameter" should be defined here.

Page 4 lines 32-33: Please specify what models of gas monitors.

Page 6 lines 25-31: It may be useful to indicate why theta is used rather than R2 and advantages/disadvantages.

Please reference the figures in order. Currently Figure 8a is referenced after Figure 1 and before the others (page 7).

Table 1: Please include the total length of each experiment in the table. In the caption, please indicate that the d-butanol tracer was not added in experiment 1.

Page 9 line 19: Please consider adding a figure (perhaps to the supplemental material) that show the OA mass change throughout the experiments both with and without wall loss correction.

Page 11 line 17: Given the nature of the analysis, "similarity" rather than "correlation" may be a better word choice.

Figure 5: Would it be more appropriate to show the natural log of the PTR-MS signal of butanol normalized to the initial value? Also, please remind the reader that m/z 66 is butanol in the caption.

**References**

Kroll, J. H., Lim, C. Y., Kessler, S. H. and Wilson, K. R.: Heterogeneous Oxidation of Atmospheric Organic Aerosol: Kinetics of Changes to the Amount and Oxidation State of Particle-Phase Organic Carbon, J. Phys. Chem. A, 119(44), 10767–10783, doi:10.1021/acs.jpca.5b06946, 2015.

---

## Author Response (AR2)

**Comments:**

1. I would like to see a more thorough discussion on how the chemistry in the chamber experiments would be expected to be similar to or different from that occurring in the atmosphere. In particular, I would like to see evaporation of POA, loss of gas-phase species to walls, $NO_x$ concentration, and $NO/NO_2$ partitioning to be addressed.

We have added a brief discussion about the similarities and differences of our experiments with the atmosphere. The evaporation of the POA is addressed in a separate paper by Louvaris et al. (in preparation) combining isothermal dilution and thermodenuder measurements. For the five chamber experiments the $NO_x$ concentrations were in the range from 1.5 to 8 ppb. The $NO_2$ to NO ratio ranged from 2 to above 10. There was no detectable decrease of the concentrations of the VOCs measured by the PTR-MS (e.g. loss to the chamber walls) during the characterization periods. The above information has been added to the revised paper.

2. Page 7 line 22: A change in the CE would presumably make the AMS vs. SMPS comparison worse. Please revise and expand this section to provide information on how the suggested factors would influence the comparison and what order of magnitude effect would be expected.

Applying the algorithm of Kostenidou et al (2007), which compares the AMS mass distributions and the SMPS volume distributions assuming spherical particles, we estimated an effective CE around 5. On the other hand, a CE lower than unity would make the comparison even worse. This is similar to the discrepancy between the mass concentrations estimated using the SMPS measurements (assuming spherical particles and density equal to unity) and the filter mass measurements. Explaining this discrepancy would require a density of the COA much higher than unity and the particles to be quite non-spherical. This suggests that the error introduced by assuming spherical particles is of the order of 2-4. This information has been added to the paper.

3. Page 9 lines 21-23: There is a significant body of literature regarding heterogeneous oxidation reactions (e.g., Kroll et al. (2015) and references therein). Are the results presented here consistent with previous works in terms of the O:C and H:C changes observed at the levels of oxidant exposure achieved in these experiments? Given the large body of work on heterogeneous oxidation, a more thorough discussion of this here would be prudent.

The changes observed in Kroll et al. (2015) are qualitatively consistent with our observations, but the corresponding timescales are very different. In our study the changes in both the laboratory and the field take place at OH exposures that are at least one order of magnitude lower than those required in the Kroll et al. (2015) study. However, in our experiments the particles are exposed to ozone also. The results of laboratory studies of oleic acid ozonolysis suggest that the corresponding reactions can take place in as little as minutes (e.g., Morris et al. (2002)), something consistent with the observations here. These suggest that the ozonolysis is probably the most important pathway for the observed changes and not the reaction with OH. We have added a paragraph discussing this important issue.

4. Page 10 lines 8-10: How different in terms of theta were the aged factors from each other? For the aged factor used later in the paper (particularly in Figure 11), was an average aged factor used, or one from a specific experiment? Was there a noticeable difference in between the ozonolysis only factor and those aged with OH?

The aged factors after exposure to UV (Experiments 1, 2, and 4) were similar to each other (theta ranging from 2 to 6 degrees). The corresponding angles between the dark ozonolysis experiment (Experiment 3) and the UV exposure ones were higher ranging from 8 to 14 degrees as the dark ozonolysis factor was less oxidized. The dark ozonolysis factor was between the fresh and UV-aged COA. For the results shown in Figure 11, an average spectrum was used for the fresh factor and the average UV-exposure factor was used for the aged factor. This is now explained in the paper.

5.      Page 11 line 7: Here a theta of 13 degrees is defined as "quite similar" whereas on page 6 line 28 a theta of 11-15 is discussed as having "many similarities though they are not the same" and in Sect. 3.4 a change of 15 degrees is discussed as being significant in terms of the changes observed due to ozonolysis. Likewise, the use of "significantly" on page 11 line 27 in describing a change of 15 degrees should be reconsidered. While I recognize that this analysis is somewhat qualitative, it would be beneficial to maintain more consistent descriptions throughout the manuscript.

Part of the inconsistency is due to the comparisons of mass spectra measured in the same experiments and those derived independently by PMF analysis of independent ambient datasets. We have rephrased the corresponding sentences to maintain consistency throughout the manuscript. We have deleted the word "significantly" on page 11 line 27 and just mention the corresponding theta angle.

**Technical Corrections:**

6.      Page 2 line 32: Suggest change to "..indicate that commercial and residential cooking contribute to…"

The proposed change has been included in the revised paper.

7.      Page 3 line 3: Suggest change to "…may significantly alter…"

Changed.

8.      Page 3 line 7: BC is not a primary organic aerosol component.

The sentence in Page 3, line 7 was rephrased.

9.      Page 3 line 23-24: What is the size of the chamber itself?

The proposed information has been included in the revised manuscript.

10.     Section 2.1: Please clarify if an OH precursor was used.

No OH precursor was used in any of the experiments. This has been added to section 2.1

11.     Page 4 line 8: The voltage difference between the filament and the ion chamber is 70 V.

Changed.

12.     Page 4 line 21: "P parameter" should be defined here.

A brief description of the P parameter has been added.

13.     Page 4 lines 32-33: Please specify what models of gas monitors.

The gas monitor models have been included in the revised manuscript.

14. **Page 6 lines 25-31: It may be useful to indicate why theta is used rather than R2 and advantages/ disadvantages.**

The advantage of angle theta use for mass spectra comparisons is that it can detect small differences that the correlation coefficient $R^2$ cannot. For example, small differences of 1-5 degrees all correspond to an $R^2 = 0.99$. However, if the difference is quite large (e.g. theta >30 degrees) then $R^2$ works equally well. This is now explained in the paper.

15. **Please reference the figures in order. Currently Figure 8a is referenced after Figure 1 and before the others (page 7).**

The order of the figures has been rearranged in the revised paper.

16. **Table 1: Please include the total length of each experiment in the table. In the caption, please indicate that the d-butanol tracer was not added in experiment 1.**

Table 1 has been updated with the proposed information.

17. **Page 9 line 19: Please consider adding a figure (perhaps to the supplemental material) that show the OA mass change throughout the experiments both with and without wall loss correction.**

A figure showing the OA mass change with and without the wall loss correction was added in the revised SI.

18. **Page 11 line 17: Given the nature of the analysis, "similarity" rather than "correlation" may be a better word choice.**

The proposed change has been included in the revised manuscript.

19. **Figure 5: Would it be more appropriate to show the natural log of the PTR-MS signal of butanol normalized to the initial value? Also, please remind the reader that m/z 66 is butanol in the caption.**

The caption of Figure 5 now explains that m/z 66 corresponds to d-butanol.

**References**

Kroll, J. H., Lim, C. Y., Kessler, S. H. and Wilson, K. R.: Heterogeneous oxidation of atmospheric organic aerosol: Kinetics of changes to the amount and oxidation state of particle-phase organic carbon, J. Phys. Chem. A, 119, 10767–10783, 2015.

Louvaris, E. E., E. Karnezi, E. Kostenidou, C. Kaltsonoudis, and S. N. Pandis: Estimation of the volatility distribution of organic aerosol combining thermodenuder and isothermal dilution measurements. In preparation.

Morris, J. W., Davidovits, P., Jayne, J. T., Jimenez, J. L., Shi, Q., Kolb, C. E., Worsnop, D. R., Barney, W. S., and Cass G.: Kinetics of submicron oleic acid aerosols with ozone: A novel aerosol mass spectrometric technique, Geophys. Res. Lett., doi: 10.1029/20002GL014692, 2002.